# Defining the function of OmpA in the Rcs stress response

Kilian Dekoninck[1,2†], Juliette Létoquart[1,2†], Cédric Laguri[3], Pascal Demange[4], Robin Bevernaegie[5], Jean-Pierre Simorre[3], Olivia Dehu[2], Bogdan I Iorga[2,6], Benjamin Elias[5], Seung-Hyun Cho[1,2]*, Jean-Francois Collet[1,2]*

[1]WELBIO, Brussels, Belgium; [2]de Duve Institute, Université catholique de Louvain (UCLouvain), Brussels, Belgium; [3]Univ. Grenoble Alpes, CNRS, CEA, Grenoble, France; [4]Institut de Pharmacologie et de Biologie Structurale, IPBS, Université de Toulouse, CNRS, Université Paul Sabatier, Toulouse, France; [5]Institut de la Matière Condensée et des Nanosciences (IMCN), Molecular Chemistry, Materials and Catalysis (MOST), Université catholique de Louvain (UCLouvain), Louvain-la-Neuve, Belgium; [6]Université Paris-Saclay, CNRS UPR 2301, Institut de Chimie des Substances Naturelles, Gif-sur-Yvette, France

*For correspondence:
seung.cho@uclouvain.be (S-HC);
jean-francois.collet@uclouvain.be
(J-FC)

[†]These authors contributed
equally to this work

Reviewing editor: Michael T
Laub, Massachusetts Institute of
Technology, United States

**Abstract** OmpA, a protein commonly found in the outer membrane of Gram-negative bacteria, has served as a paradigm for the study of β-barrel proteins for several decades. In *Escherichia coli*, OmpA was previously reported to form complexes with RcsF, a surface-exposed lipoprotein that triggers the Rcs stress response when damage occurs in the outer membrane and the peptidoglycan. How OmpA interacts with RcsF and whether this interaction allows RcsF to reach the surface has remained unclear. Here, we integrated in vivo and in vitro approaches to establish that RcsF interacts with the C-terminal, periplasmic domain of OmpA, not with the N-terminal β-barrel, thus implying that RcsF does not reach the bacterial surface via OmpA. Our results suggest a novel function for OmpA in the cell envelope: OmpA competes with the inner membrane protein IgaA, the downstream Rcs component, for RcsF binding across the periplasm, thereby regulating the Rcs response.

## Introduction

The cell envelope is the morphological hallmark of Gram-negative bacteria. It consists of an inner membrane (IM) surrounding the cytoplasm as well as an outer membrane (OM), an asymmetric bilayer with phospholipids in the inner leaflet and lipopolysaccharides in the outer leaflet (*Silhavy et al., 2010*). The two membranes are separated by the periplasm, a compartment in which lies a thin layer of peptidoglycan. The cell envelope is essential for viability: the OM serves as a permeability barrier against toxic compounds present in the environment while the peptidoglycan provides shape and osmotic protection to cells (*Okuda et al., 2016*; *Typas et al., 2012*; *Egan et al., 2020*).

Given the functional and structural importance of the envelope, bacteria need to respond to breaches in envelope integrity in a fast and adequate manner. Bacteria have therefore evolved sophisticated signaling systems that monitor envelope integrity and respond to perturbations (*Ruiz and Silhavy, 2005*; *MacRitchie et al., 2008*; *Delhaye et al., 2019*). In *Escherichia coli* and the Enterobacteriaceae, the Rcs system detects damage to the OM and the peptidoglycan (*Wall et al., 2018*; *Laubacher and Ades, 2008*; *Farris et al., 2010*). In response, Rcs modulates the expression of dozens of genes, including those involved in the biosynthesis of colanic acid, an exopolysaccharide that accumulates on the cell surface to form a protective capsule (*Wall et al., 2018*; *Laloux and Collet, 2017*).

Rcs signal transduction involves a multi-step phosphorelay (*Wall et al., 2018*). Under stress, the IM histidine kinase RcsC autophosphorylates, transfers the phosphoryl group to the IM protein RcsD and finally to the cytoplasmic response regulator RcsB. Rcs activity is modulated by two proteins that are not part of the phosphorylation cascade: RcsF and IgaA. RcsF is an OM lipoprotein that senses most Rcs-inducing cues, while IgaA is an essential IM protein that down-regulates Rcs (*Takeda et al., 2001*; *Domínguez-Bernal et al., 2004*) by interacting with RcsD (*Wall et al., 2020*). When perturbations occur in the peptidoglycan or in the OM, RcsF, while remaining anchored in the OM, reaches across the periplasm to interact with IgaA, leading this protein to alleviate its inhibition of the phosphorelay, turning on Rcs (*Cho et al., 2014*; *Hussein et al., 2018*). In the absence of stress, RcsF is occluded from IgaA by interacting with OM proteins. A complex between RcsF and BamA, the core component of the β-barrel assembly machinery (BAM), was identified (*Cho et al., 2014*; *Konovalova et al., 2014*) and its structure solved (*Rodríguez-Alonso et al., 2020*). This complex forms as an intermediate (*Cho et al., 2014*; *Konovalova et al., 2014*): delivery of unfolded OM β-barrels (OMPs) to BAM triggers the release of RcsF from BamA and its transfer to OMP partners (*Rodríguez-Alonso et al., 2020*). Three abundant OMPs (OmpA, OmpC and OmpF) have been identified as RcsF partners (*Cho et al., 2014*; *Konovalova et al., 2014*). Under stress conditions, newly synthesized RcsF molecules fail to interact with BamA (*Cho et al., 2014*): they remain in the periplasm, free to bind IgaA, triggering Rcs.

Crucially, whereas the general view is that OM lipoproteins are oriented toward the periplasm, previous work concluded that at least a portion of RcsF, a protein which is composed of an N-terminal disordered linker and a C-terminal globular domain required for signaling (*Leverrier et al., 2011*; *Rogov et al., 2011*), is exposed on the cell surface; OmpA, OmpC, and OmpF, but not BamA (*Cho et al., 2014*), were identified as potential vehicles for RcsF surface exposure (*Cho et al., 2014*; *Konovalova et al., 2014*; *Konovalova et al., 2016*). In a topological model of RcsF surface exposure (*Konovalova et al., 2014*), the lipid moiety of RcsF is anchored in the outer leaflet of the OM and the N-terminal disordered linker is exposed on the cell surface before being threaded through the lumen of the OMP partners. However, definitive evidence for this model is still lacking.

In addition, because OmpC, OmpF, and OmpA belong to two distinct structural groups, it is unclear whether RcsF interacts with its three OMP partners in a similar way. Indeed, OmpC and OmpF form 16-stranded β-barrels that associate into trimers in the OM. Because they form large β-barrels, they display a central pore (*Baslé et al., 2006*; *Yamashita et al., 2008*; *Radhakrishnan et al., 2010*; *Housden et al., 2013*) that is large enough to accommodate a disordered peptide such as the RcsF linker. The situation is less clear for OmpA: although this protein, the most abundant OMP in *Escherichia coli*, has been studied for more than four decades, how it folds remains controversial. While some studies indicate that OmpA can also fold into a 16-stranded β-barrel with a large central pore (*Singh et al., 2003*; *Stathopoulos, 1996*), the predominant view is that OmpA adopts a two-domain structure with an N-terminal eight-stranded β-barrel inserted in the OM (*Pautsch and Schulz, 1998*) and a C-terminal, globular domain in the periplasm (*Mot and Vanderleyden, 1994*; *Park et al., 2012*). In this conformation, the β-barrel of OmpA is too small to accommodate a polypeptide. Thus, despite the tremendous work that has been done on OmpA, we do not know whether this protein adopts the large β-barrel structure when in complex with RcsF, or whether it folds into the predominant two-domain conformation. If the latter, where does RcsF bind OmpA?

To resolve these outstanding structural and mechanistic questions, here we dissected the OmpA-RcsF complex. By combining in vivo site-specific photo-crosslinking, targeted proteolysis, and nuclear magnetic resonance (NMR) titration, we established that OmpA adopts its two-domain structure when in complex with RcsF and that it is the C-terminal, periplasmic domain—not the β-barrel—that interacts with the lipoprotein. In addition, we identified residues in RcsF and in OmpA that are involved in the interaction, thus providing information about the binding interface. Taken together, our results indicate that the topology of OmpA-RcsF is different from that of OmpC/F-RcsF; they also imply that RcsF does not use OmpA to reach the cell surface. This has important implications for how RcsF senses OM stress: if the linker of RcsF is not on the surface in the OmpA-RcsF complex, then OmpA-RcsF cannot serve to monitor the state of the lipopolysaccharide leaflet via direct interactions with lipopolysaccharide molecules, as previously proposed (*Konovalova et al., 2016*). Finally, we determined the equilibrium dissociation constants of both the C-terminal domain of OmpA and the periplasmic domain of IgaA for RcsF and provide evidence suggesting that OmpA

and IgaA compete for RcsF binding across the periplasm. Our results support a model in which OmpA serves as a buffer for RcsF, titrating it from IgaA, thereby fine-tuning Rcs activity.

## Results

### RcsF interacts with the C-terminal region of OmpA in vivo

The stress sensor lipoprotein RcsF was previously shown to be surface-exposed (*Cho et al., 2014*; *Konovalova et al., 2014*), and OmpA was described as a possible vehicle for its surface exposure (*Cho et al., 2014*; *Konovalova et al., 2014*). However, how OmpA folds when in complex with RcsF and whether this interaction allows RcsF to become surface-exposed remain to be determined.

To close this gap, we characterized the OmpA-RcsF interaction. In a previous study, we identified six RcsF residues (in the N-terminal disordered linker and at the tip of the signaling domain) as being part of the interaction interface between RcsF and OmpA (*Cho et al., 2014*). These residues were identified using a site-specific photo-crosslinking strategy in which a photoreactive, crosslinkable amino acid is inserted at specific positions in the protein of interest, with the help of an exogenous orthogonal tRNA/aminoacyl-tRNA synthetase pair (*Chin et al., 2002*). We first sought to confirm and extend these results to more clearly define the binding interface in RcsF. Instead of using the hydrophobic crosslinker p-benzoyl-L-phenylalanine like before (*Cho et al., 2014*), we used $N^6$-((3-(3-methyl-3H-diazirin-3-yl)propyl)carbamoyl)-L-lysine (DiZPK), a lysine analog with substantially higher photo-crosslinking efficiency than p-benzoyl-L-phenylalanine (*Zhang et al., 2011*). We selected 11 positions distributed along the RcsF sequence, including four (R21, Q28, Q33, R45) in the disordered linker and seven on the surface of the signaling domain (N54, Q79, R89, K98, E110, P116, Q121) (*Figure 1*). After UV illumination, a ~ 55 kDa band, the size of the OmpA (40 kDa)-RcsF (14 kDa) complex, formed and was detected with an anti-RcsF antibody (*Figure 1*) for the following variants (in decreasing intensity): RcsF$_{R89X}$, RcsF$_{P116X}$, RcsF$_{Q79X}$, RcsF$_{R45X}$, RcsF$_{K98X}$, RcsF$_{Q121X}$, RcsF$_{N54X}$, and RcsF$_{E110X}$. The identity of the ~55 kDa band as OmpA-RcsF was further verified by showing that it did not form in Δ*ompA* cells (*Figure 1—figure supplements 1* and *2*). Thus, as in our previous study (*Cho et al., 2014*), complex formation was observed when the photoactivatable amino acid was inserted in the linker (R45) or at the tip of the signaling domain (Q79 and P116). The complex also formed when DiZPK was incorporated in other regions of the signaling domain, such as α-helix 1 (N54), α-helix 2 (R89 and K98), β-strand 2 (E110), and β-strand 3 (Q121) (*Figure 1*). Noteworthy, residues Q79, R89 and P116 are part of the binding interface between RcsF and the luminal wall of the BamA β-barrel in the recently published structure of the BamA-RcsF complex (*Rodríguez-Alonso et al., 2020*). Taken together, these observations substantially enlarge the region of RcsF known to interact with OmpA. Of note, we observed that the RcsF$_{R45X}$, RcsF$_{R89X}$ and RcsF$_{K98X}$ variants formed a UV-dependent band migrating slightly higher than the OmpA-RcsF complex (*Figure 1*). Focusing on RcsF$_{R89X}$, we identified this band as a complex between RcsF and OmpC/F because it did not form when *ompR*, a transcription factor required for the production of OmpC and OmpF (*Hall and Silhavy, 1979*; *Chubiz and Rao, 2011*), was deleted (*Figure 1—figure supplement 2*).

We next sought to identify where RcsF binds OmpA. The general view is that OmpA consists of two domains, an eight-stranded β-barrel anchoring the protein in the OM and a soluble C-terminal domain located in the periplasm, where it binds the peptidoglycan (*Mot and Vanderleyden, 1994*; *Park et al., 2012*; *Figure 2A*). However, an alternative conformation has been proposed in which OmpA folds into a large, 16-stranded β-barrel (*Singh et al., 2003*; *Stathopoulos, 1996*; *Figure 2A*). To gain insight into where RcsF binds OmpA and to characterize OmpA's conformation when in complex with RcsF, we first engineered an OmpA variant with a thrombin cleavage site inserted after residue V189 (OmpA$_{TH\_189}$), in the middle of the OmpA sequence (*Figure 2B*). Taking the two-domain structure as a reference, the cleavage site was inserted between the N- and C-terminal domains. We then selected three DiZPK-containing RcsF mutants (RcsF$_{R45X}$, RcsF$_{Q79X}$, RcsF$_{R89X}$) that formed a covalent complex with OmpA at high levels when exposed to UV light (*Figure 1*). These variants display DiZPK in three regions of RcsF: in RcsF$_{R45X}$, DiZPK is present at the end of the disordered linker, while RcsF$_{Q79X}$ displays DiZPK in the large loop at the tip of the signaling domain and RcsF$_{R89X}$ displays it on the central α-helix 2 (*Figure 1*). These variants were expressed in *E. coli* cells also producing OmpA$_{TH\_189}$, and complex formation was induced with UV light (*Figure 2C*). For all

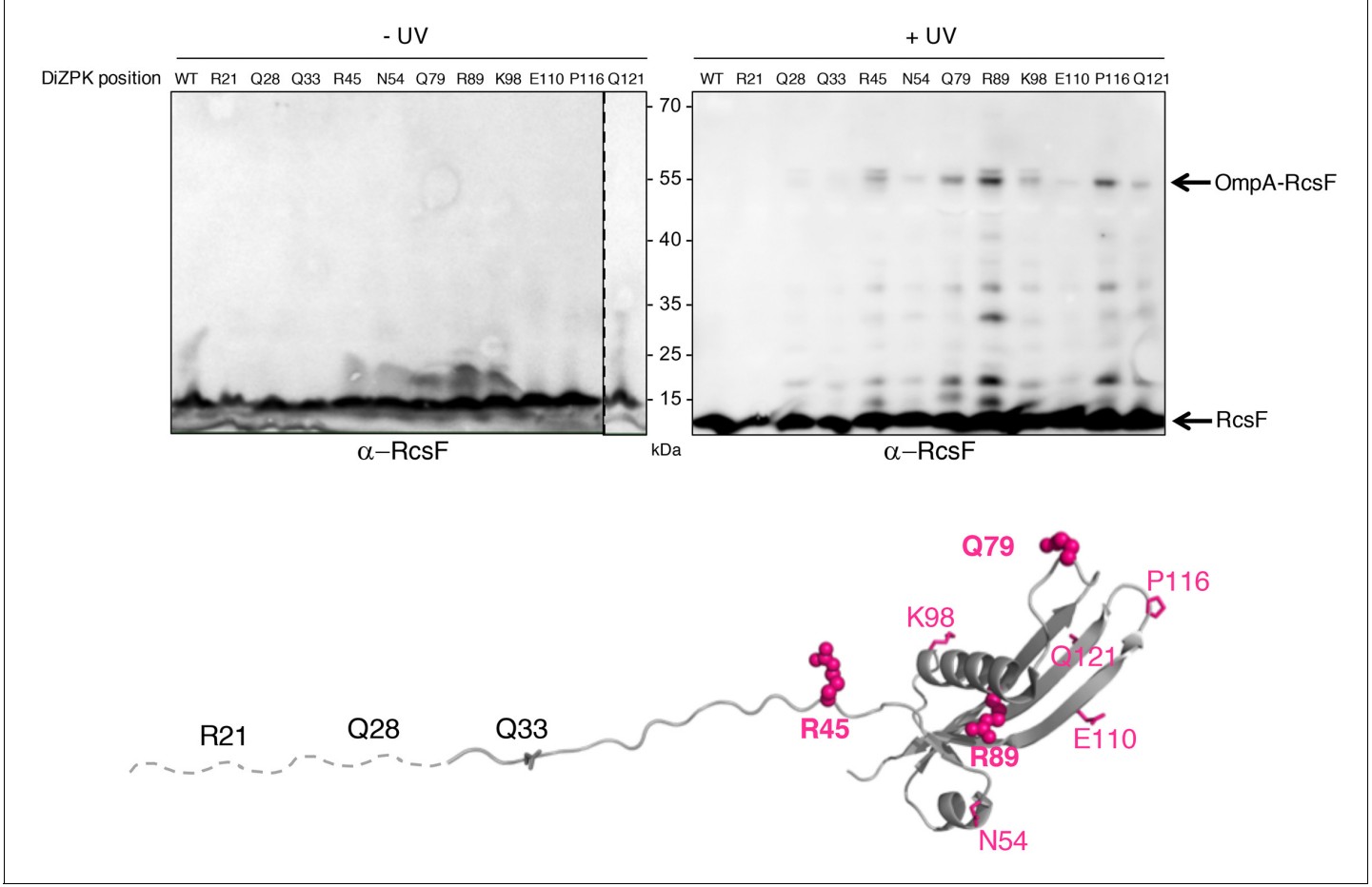

**Figure 1.** Defining the binding interface of RcsF on OmpA using in vivo site-specific photo-crosslinking. Upper panel: Δ*rcsF* cells expressing wild-type (WT) RcsF or DiZPK-containing RcsF variants (from pSC253) were irradiated with UV light (+) or not (-), and protein samples were immunoblotted with an anti-RcsF antibody. A 55 kDa band, corresponding to the size of the OmpA-RcsF complex, was observed for eight of the mutants (R45, N54, Q79, R89, K98, E110, P116, Q121). Lower panel: residues of RcsF were replaced by DiZPK to map the zone of interaction with OmpA. In this cartoon of the NMR structure of RcsF (PDB: 2L8Y), the truncated N-terminal portion of the protein is shown as a dashed line and the residues that were found to interact with OmpA appear in magenta. The side chains of the residues that were selected for further experiments are shown as spheres, and other side chains are represented as sticks.

The online version of this article includes the following figure supplement(s) for figure 1:

**Figure supplement 1.** The UV-dependent 55 kDa band is the OmpA-RcsF complex.

**Figure supplement 2.** The band migrating above the OmpA-RcsF complex is the OmpC/F-RcsF complex.

three RcsF variants, digestion of the 55 kDa OmpA$_{TH\_189}$-RcsF complex with thrombin yielded a ~ 35 kDa band that was recognized by both anti-RcsF and anti-His antibodies (*Figure 2C,D*). Given the presence of a His-tag in the C-terminus of OmpA (Materials and Methods), we concluded that RcsF (14 kDa) interacts with the C-terminal region of OmpA (~16 kDa) in vivo.

To further delineate the region of OmpA that is important for complex formation, a second OmpA mutant with a thrombin cleavage site inserted after residue I243 (OmpA$_{TH\_243}$) was generated (*Figure 2B*). Like OmpA$_{TH\_189}$, OmpA$_{TH\_243}$ (~10 kDa) could be photo-crosslinked to the three DiZPK-containing RcsF variants described above (*Figure 2C,D*). In all three cases, thrombin digestion of the OmpA$_{TH\_243}$-RcsF complexes generated a smaller band (~30 kDa) that was recognized by anti-RcsF antibodies (*Figure 2C,D*). In contrast to OmpA$_{TH\_189}$-RcsF, which underwent complete digestion, OmpA$_{TH\_243}$-RcsF only underwent partial cleavage (*Figure 2C,D*), which probably reflected decreased accessibility of the cleavage site to the protease. This ~30 kDa band was also detected by anti-His antibodies (*Figure 2D*), indicating that the three tested residues of RcsF bind the region of OmpA between residue I243 and the C-terminus.

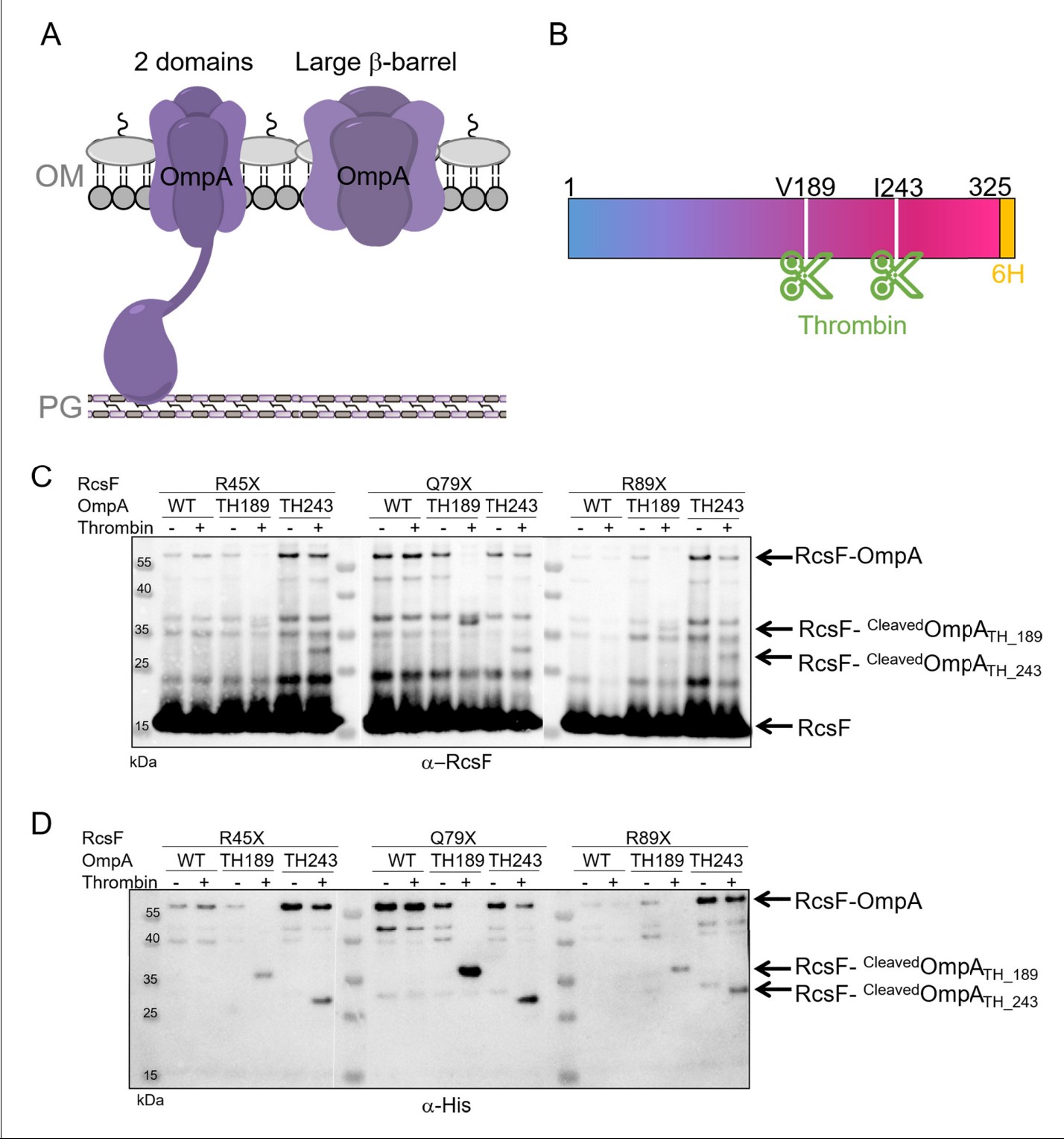

**Figure 2.** RcsF interacts with the C-terminal portion of OmpA in vivo. (**A**) Schematic of the two conformations of OmpA. Left: the predominant view is that OmpA adopts a two-domain structure, with an N-terminal β-barrel embedded in the OM and a C-terminal periplasmic domain binding the peptidoglycan. Right: an alternative conformation in which OmpA folds into a large β-barrel has also been proposed (*Singh et al., 2003*; *Stathopoulos, 1996*). (**B**) Schematic of OmpA variants containing a thrombin-specific cleavage site (scissors at positions V189 and I243) and a 6x-histidine tag (orange) at the C-terminus. (**C–D**) In vivo site-specific photo-crosslinking of RcsF. Δ*rcsF* cells co-expressing one of the DiZPK-containing RcsF variant (R45X, Q79X, or R89X) together with OmpA (wild-type, or with a thrombin site inserted at V189 or I243) were UV-irradiated. The RcsF

*Figure 2 continued on next page*

*Figure 2 continued*

variants were expressed from pSC253; OmpA, OmpA$_{TH\_189}$, and OmpA$_{TH\_243}$ were expressed from the chromosome. After immunoprecipitation with anti-RcsF, protein samples were incubated (+) or not (-) with thrombin and immunoblotted with an anti-RcsF (**C**) or an anti-His-tag antibody (**D**). At least partial digestion of the ~55 kDa complex corresponding to OmpA-RcsF occurred with all three DiZPK-containing RcsF variants, yielding a band (RcsF-$^{Cleaved}$OmpA$_{TH\_189}$ or RcsF-$^{cleaved}$OmpA$_{TH\_243}$) migrating at lower molecular weights that was detected by both antibodies.

## The C-terminal region of OmpA is necessary and sufficient for binding RcsF

The results above indicated that RcsF interacts with the C-terminal region of OmpA, raising the question of whether the N-terminal region also participates in this interaction. To probe this potential interaction directly, we generated an OmpA variant lacking the C-terminal moiety (OmpA$_{1-170}$) and tested whether it could be crosslinked to RcsF (*Figure 3A*). Chemical crosslinking was carried out with 3,3'-dithio-bis[sulfosuccinimidylpropionate] (DTSSP) (*Cho et al., 2014*). Although complexes formed in cells expressing wild-type OmpA, no complex was detected in cells producing OmpA$_{1-170}$ (*Figure 3A*). In addition, expression of OmpA$_{1-170}$ did not suppress the activation of the Rcs system (*Figure 3A*, see *Figure 3—source data 1* for statistics) that occurs in cells lacking OmpA and that results from the inability of RcsF to interact with this β-barrel (*Cho et al., 2014*).

To test whether the N-terminal domain of OmpA was indirectly required for the OmpA-RcsF complex to form, we generated a hybrid protein (OmpA$_X$) in which the C-terminal region of OmpA (OmpA$_{171-325}$), corresponding to the periplasmic domain in the two-domain structure, was fused to OmpX (*Figure 3B*). OmpX is a small eight-stranded β-barrel that constitutes a structural homolog of the N-terminal region of OmpA when it folds as a small β-barrel (the two β-barrels can be superimposed with a root mean square deviation (RMSD) of 2.49 Å; *Figure 3—figure supplement 1A*). OmpX does not share sequence homology with the N-terminal region of OmpA (~25%) and can thus be considered as an OM anchor for the C-terminal region when fused to the latter as in OmpA$_X$ (*Figure 3B*). We found that OmpA$_X$ could be crosslinked to RcsF (*Figure 3B*) and that its expression fully suppressed Rcs activation (*Figure 3B*, see *Figure 3—source data 1* for statistics). Because we could not completely exclude the unlikely possibility that OmpA$_X$ could rearrange into a large β-barrel able to bind RcsF, we prepared an additional variant of OmpA (OmpA$_{Pal}$) in which the C-terminal domain (OmpA$_{171-325}$) was fused to the signal sequence and lipobox (for lipid modification; *Szewczyk and Collet, 2016*) of the OM lipoprotein Pal, thus converting the C-terminal domain of OmpA into a lipoprotein (*Figure 3C*). Remarkably, expression of OmpA$_{Pal}$ led to the formation of OmpA$_{Pal}$-RcsF and substantially decreased Rcs activity (*Figure 3C*, see *Figure 3—source data 1* for statistics). In these experiments, we confirmed that the expression levels of OmpA$_{1-170}$, OmpA$_X$ and OmpA$_{Pal}$ were similar to those of the wild type (*Figure 3—figure supplement 2*). Thus, the C-terminal region of OmpA is necessary and sufficient for the interaction with RcsF, and the N-terminal region is dispensable. Altogether, these results are consistent with the conclusion that OmpA adopts its two-domain structure—and not the large β-barrel conformer—when in complex with RcsF. We therefore conclude that RcsF interacts with the C-terminal, globular domain of OmpA and that this interaction takes place on the periplasmic side of the OM.

## RcsF interacts with the periplasmic domain of OmpA in vitro

We next sought to validate our in vivo observations by probing the formation of a complex between RcsF and the soluble, periplasmic domain of OmpA (OmpA$_{186-325}$) in vitro using purified proteins. However, attempts to pull-down OmpA$_{186-325}$ with a soluble, His-tagged version of RcsF failed (data not shown), suggesting that the interaction between these two proteins was weak. Because NMR is a highly effective tool to investigate weak protein-protein interactions (*Vaynberg and Qin, 2006*), we employed NMR titration experiments of $^{15}$N-labeled OmpA$_{186-325}$ by RcsF. In this approach, the $^{15}$N-$^{1}$H 2D NMR spectra of OmpA were recorded upon addition of increasing amounts of RcsF (2 and 10 molar equivalents); when OmpA and RcsF interact, concentration-dependent perturbations in the NMR spectra appear. Upon addition of RcsF, several OmpA residues showed chemical shift variations in $^{15}$N-$^{1}$H 2D correlation experiments (*Figure 4A,B*, see *Figure 4—source data 1* for details). Most of the shifted residues (T240, G244, S245, D246, A247, G251, L252, K294 and A297; *Figure 4B*) were located near the tip of the periplasmic domain (*Figure 4C,D*), identifying this

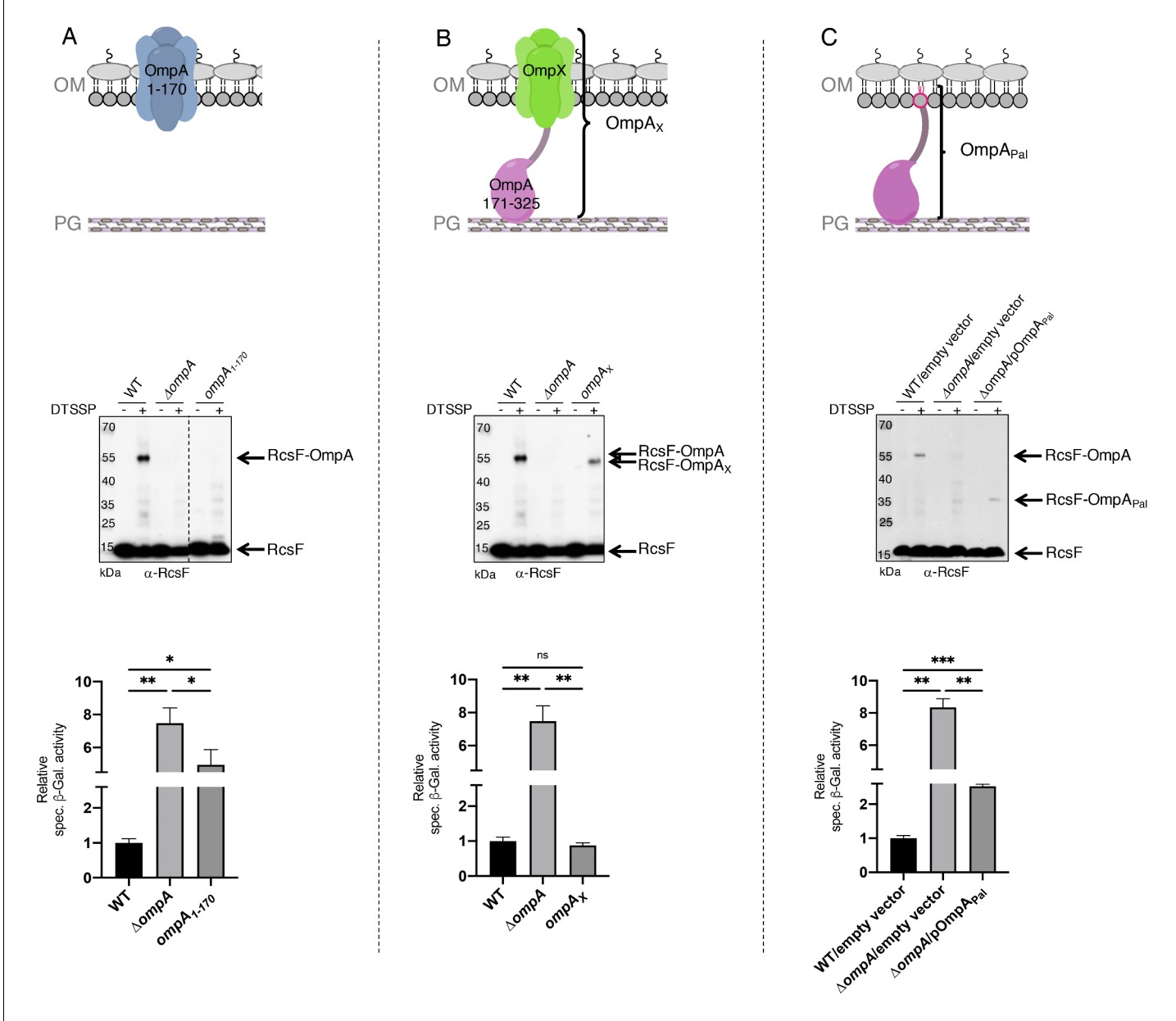

**Figure 3.** The periplasmic domain of OmpA is necessary and sufficient for the interaction with RcsF in vivo. (**A**) *Upper panel:* Schematic of the truncated OmpA variant corresponding to the 8-stranded ß-barrel domain (OmpA$_{1-170}$). *Middle panel:* In vivo chemical crosslinking of RcsF to OmpA and OmpA$_{1-170}$. WT, Δ*ompA* and Δ*ompA*::*ompA$_{1-170}$* DH300 cells were incubated with or without 3,3'-dithio-bis[sulfosuccinimidylpropionate] (DTSSP). Proteins were immunoblotted with anti-RcsF (same for the middle panel in **B** and **C**). The OmpA-RcsF complex was detected in WT cells but not in cells expressing OmpA$_{1-170}$. *Lower panel:* ß-galactosidase (ß-gal) activity was measured using the transcriptional *rprA-lacZ* fusion on the chromosome (**Majdalani et al., 2002**) using the same DH300 strains as in the middle panel (same for the lower panel in **B** and **C**). Deleting *ompA* induces Rcs, expression of OmpA$_{1-170}$ does not restore basal Rcs activity. (**B**) *Upper panel:* Schematic of the hybrid protein consisting of a fusion between the 8-stranded ß-barrel OmpX and the periplasmic domain of OmpA (OmpA$_{171-325}$) (OmpA$_X$). *Middle panel:* In vivo chemical crosslinking of RcsF to OmpA and OmpA$_X$. WT, Δ*ompA* and Δ*ompA*::*ompA$_X$* cells were incubated with or without DTSSP. The RcsF-OmpA complex was detected in WT cells as well as in cells expressing OmpA$_X$ (RcsF-OmpA$_X$). *Lower panel:* Deleting *ompA* induces Rcs, expression of OmpA$_X$ restores basal Rcs activity. (**C**) *Upper panel:* Schematic of the C-terminal domain of OmpA (OmpA$_{171-325}$) fused to the signal sequence and lipobox of the OM lipoprotein Pal (OmpA$_{Pal}$). *Middle panel:* In vivo chemical crosslinking of RcsF to OmpA and OmpA$_{Pal}$. WT and Δ*ompA* harboring pDSW204, an empty vector, used as control, and Δ*ompA* cells harboring pKiD22, expressing OmpA$_{Pal}$ from an IPTG-inducible promoter, were incubated with or without DTSSP. The OmpA-RcsF complex was detected in WT cells as well as in cells expressing OmpA$_{Pal}$ (OmpA$_{Pal}$-RcsF). *Lower panel:* Deleting *ompA* induces Rcs, expression of OmpA$_{Pal}$ substantially decreases Rcs activity. See *Figure 3—source data 1* for details and statistics of middle panels. Mean (n = 3) and standard deviation (error bars) are shown. Differences were evaluated with Student's *t* test (ns, not significant; *p<0.05; ***p<0.001).

*Figure 3 continued on next page*

*Figure 3 continued*

The online version of this article includes the following source data and figure supplement(s) for figure 3:

**Source data 1.** Raw source data for middle panels of *Figure 3A,B,C*.
**Figure supplement 1.** The eight-stranded ß-barrels of OmpA and OmpX are structurally similar and do not display an open channel through the membrane.
**Figure supplement 2.** The expression levels of the OmpA variants are similar to those of wild-type OmpA.

region, and in particular a flexible loop between β-strand two and α-helix 3, as part of the binding interface with RcsF. The importance of this loop for the OmpA-RcsF interaction was confirmed using site-specific photo-crosslinking: OmpA was strongly crosslinked to RcsF when DiZPK was introduced at residue D246, while weak complex formation was observed with $OmpA_{R242X}$ and $OmpA_{Y248X}$ (*Figure 4E*). These results nicely fit with those obtained using $OmpA_{TH\_243}$ (*Figure 2B*) that identified the same region of the C-terminal domain of OmpA as part of the zone of interaction with RcsF. Thus, taken together, our results allow us to conclude that RcsF interacts with the C-terminal domain of OmpA in its globular conformation, not only in vivo but also in vitro.

## IgaA and OmpA likely compete for RcsF across the periplasmic space

RcsF turns on the Rcs response by interacting with the periplasmic domain of the IM protein IgaA under stress (*Hussein et al., 2018*). Here, we found that the interaction between RcsF and OmpA takes place in the periplasm (*Figure 5A*), thus suggesting that OmpA and IgaA compete for binding RcsF across this compartment. To investigate this hypothesis, we determined the effect of artificially increasing the IgaA concentration on the formation of the IgaA-RcsF and OmpA-RcsF complexes. In these experiments, a triple Flag-tagged, functional version of IgaA (*Hussein et al., 2018*) was expressed from an inducible plasmid. To monitor complex formation, we carried out chemical cross-linking using bis(sulfosuccinimidyl)suberate (BS3), a bifunctional crosslinker. Increasing the levels of IgaA led to changes in the IgaA-RcsF and OmpA-RcsF complexes that were inversely correlated: whereas overexpressing IgaA led to more IgaA-RcsF, it decreased the levels of the OmpA-RcsF complex (compare lanes 9 and 10 with lanes 7 and 8 in *Figure 5B*). Thus, IgaA and OmpA seem to compete for RcsF across the periplasm. Because IgaA (200 copies per cell; *Li et al., 2014*) is far less abundant than OmpA (200,000 copies), even when overexpressed (we estimate that IgaA levels were increased 8–40 fold over baseline (*Figure 5—figure supplement 1*; Materials and methods) in the experiment above), these results suggested that IgaA has a substantially higher affinity for RcsF than OmpA. To probe this directly, we determined the affinity constants of the periplasmic domains of OmpA and IgaA for RcsF. Note that the periplasmic domain of IgaA interacts with RcsF in vivo and in vitro (*Cho et al., 2014*; *Hussein et al., 2018*). First, from the NMR shift data, we calculated the equilibrium dissociation constant ($K_D$) of $OmpA_{186-325}$ for RcsF as being 125 ± 85 µM (*Figure 5C*). Second, using biolayer interferometry, we measured that the periplasmic domain of IgaA has a $K_D$ of 1.6 ± 0.3 nM for RcsF (*Figure 5D*, see *Figure 5—source data 2* for statistics). Thus, these values confirm that IgaA has substantially more affinity for RcsF than OmpA (see Discussion). Interestingly, we noted that the BamA-RcsF complex was not modified by the increased expression of IgaA (lanes 7–10 in *Figure 5B*), which is consistent with the fact that BamA has a much higher affinity for RcsF ($K_D$ ~400 nM; *Rodríguez-Alonso et al., 2020*) than OmpA.

## Discussion

### OmpA is unlikely the vehicle allowing RcsF to reach the surface

OmpA was first purified from *E. coli* membranes in 1977 (*Chai and Foulds, 1977*) and has served as a model for OMP assembly since then. Although the predominant view is that OmpA folds into a two-domain conformation, with an N-terminal eight-stranded β-barrel and a C-terminal periplasmic domain, an alternative conformation in which OmpA forms a single, 16-stranded β-barrel, has been proposed to also exist (*Singh et al., 2003*; *Stathopoulos, 1996*). Here, we integrated in vivo (*Figures 2*, *3*, *4* and *5*) and in vitro (*Figures 4* and *5*) approaches to dissect the interaction between OmpA and the lipoprotein RcsF; altogether, our data establish that OmpA is in the two-domain conformation in the OmpA-RcsF complex and that it is the C-terminal, periplasmic domain of OmpA

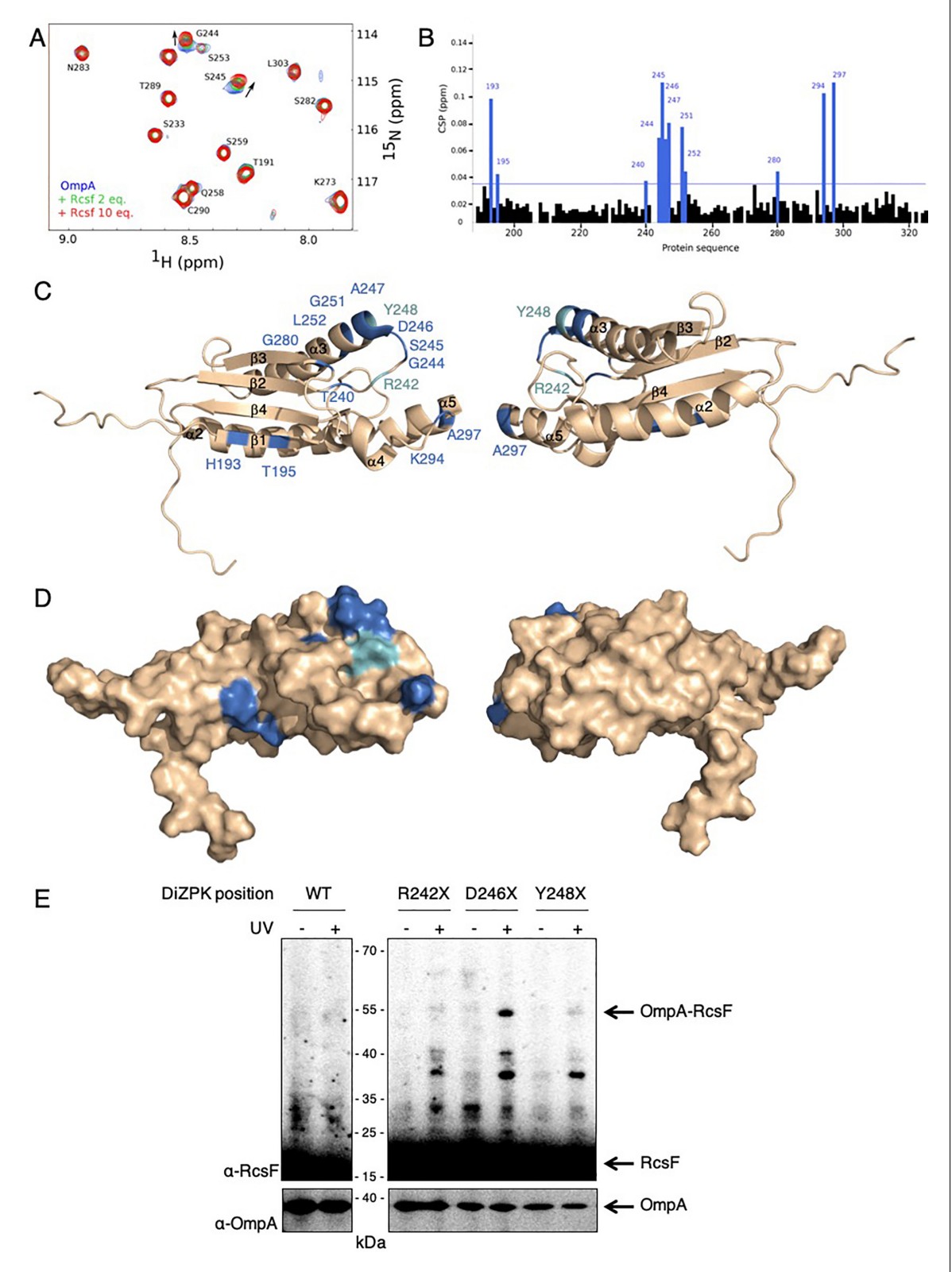

**Figure 4.** RcsF interacts with the periplasmic domain of OmpA in vitro. (**A**) An expanded region of the HSQC titration spectra of $^{15}$N-labeled OmpA$_{186-325}$ with RcsF. Several residues in the $^{1}$H/$^{15}$N BEST-TROSY-HSQC spectrum of OmpA (blue) show chemical shift perturbations (CSP) upon addition of RcsF at molar ratios of RcsF to OmpA of 2 (green) and 10 (red). Arrows indicate the direction of the chemical shift changes upon addition of RcsF to OmpA. (**B**) CSPs induced by the addition of RcsF to $^{15}$N-labeled OmpA$_{186-325}$. Residues that showed CSP larger than two standard deviation are

*Figure 4 continued on next page*

*Figure 4 continued*

colored in blue. Cartoon (C) and surface (D) view of OmpA$_{186-325}$ (PDB: 2MQE): top view (on the left) and bottom view (on the right). The residues in OmpA that undergo a CSP larger than two standard deviation (as in panel B) appear as dark blue and are labeled. Most of these residues are located between β-strand 2 and α-helix 3. The residues R242 and Y248 that do not interact well with RcsF in (E) are colored in light blue. (E) To confirm the importance of the loop between β-strand 2 and α-helix 3 of OmpA for the interaction with RcsF, we used site-specific photo-crosslinking. Δ*ompA rcsF$^+$* cells expressing wild-type (WT) OmpA or three DiZPK-containing OmpA variants (OmpA$_{R242X}$, OmpA$_{D246X}$, and OmpA$_{Y248X}$) from pPR21 were irradiated with UV light (+) or not (-), and protein samples were immunoblotted with an anti-RcsF antibody. A strong 55 kDa band, corresponding to the size of the OmpA-RcsF complex, was observed with the OmpA$_{D246X}$ variant, confirming the NMR data. Weak complex formation was also observed with OmpA$_{R242X}$ and OmpA$_{Y248X}$. The expression levels of the OmpA variants were verified by immunoblotting (lower panel).

The online version of this article includes the following source data for figure 4:

**Source data 1.** Raw source data for *Figure 4A,B*: the HSQC titration spectra of $^{15}$N-labeled OmpA$_{186-325}$ with RcsF.

that interacts with RcsF. Using protein-protein docking and molecular dynamics simulations, we built a three-dimensional model taking into accounts the results of the cross-linking experiments; in this model, RcsF and the periplasmic domain of OmpA show a good surface complementarity, with a buried accessible surface of 1087 Å$^2$. In this model, all six residues mutated to DiZPK (RcsF$_{R45X}$, RcsF$_{Q79X}$, RcsF$_{R89X}$, OmpA$_{R242X}$, OmpA$_{D246X}$, and OmpA$_{Y248X}$) are situated in close proximity to residues from the binding partner (*Figure 6A*), in good agreement with the crosslinking data. Interestingly, the peptidoglycan-binding region of OmpA (*Park et al., 2012*) remains accessible and is oriented in the direction opposite to the N-terminal end of RcsF. Our conclusions have two important implications.

First, we and others previously showed that portions of RcsF are surface-exposed (*Cho et al., 2014*; *Konovalova et al., 2014*) and proposed OmpA, OmpC, and OmpF as vehicles for surface exposure (*Cho et al., 2014*; *Konovalova et al., 2014*). Understanding how RcsF reaches the surface is crucial: lipoprotein surface exposure is an emerging concept in *E. coli* and Enterobacteriaceae (*Szewczyk and Collet, 2016*; *Konovalova and Silhavy, 2015*) and how lipoproteins cross the OM remains to be clearly established. It was proposed that it is the N-terminal linker of RcsF that is exposed on the surface before being threaded through the lumen of OmpA and other OMPs (*Konovalova et al., 2014*). However, when OmpA adopts the two-domain conformation, as in the OmpA-RcsF complex, its β-barrel domain does not have an open channel (*Figure 3—figure supplement 1B*) to accommodate the RcsF linker. We therefore conclude that RcsF does not reach the surface when in complex with OmpA, and predict that only OmpC and OmpF (which form large β-barrels) serve as vehicles for surface exposure. The fact that only a subset of the DiZPK-containing variants of RcsF that form a complex with OmpA also form a complex with OmpC/F (*Figure 1*) also supports the idea that the topology of OmpA-RcsF is different from that of OmpC/F-RcsF.

Second, a model was proposed in which RcsF, when in complex with OmpA, uses its positively charged, surface-exposed N-terminal linker to sense when interactions between lipopolysaccharide molecules are disturbed (*Konovalova et al., 2016*). However, if the RcsF linker is not surface-exposed in the OmpA-RcsF complex, as we show here, then the function of OmpA-RcsF in Rcs needs to be re-evaluated (see below). Importantly, we found that lipopolysaccharide alterations caused by sub-lethal concentrations of polymyxin B induce Rcs in MG1655 cells lacking *ompA* (*Figure 6—figure supplement 1*, see *Figure 6—figure supplement 1—source data 1* for statistics), in contrast to what was previously reported in another strain background (MC4100) (*Konovalova et al., 2016*), which further questions the potential role of OmpA-RcsF in sensing lipopolysaccharide defects.

It will also be interesting to investigate the role of the BAM machinery in the assembly of OmpA-RcsF. It was reported that the formation of the OmpA-RcsF complex was likely mediated by BAM during the assembly of OmpA (*Cho et al., 2014*; *Konovalova et al., 2014*). However, our finding that RcsF interacts with the C-terminal, periplasmic domain of OmpA, whose folding is unlikely to depend on BAM (*Noinaj et al., 2017*), questions this conclusion. Further experiments, including pulse-chase experiments, will be carried out to clarify this point. Given that BAM both interacts with RcsF and assembles OmpA in the OM, we anticipate that interpreting the data will be challenging, and therefore these experiments are outside the scope of this publication.

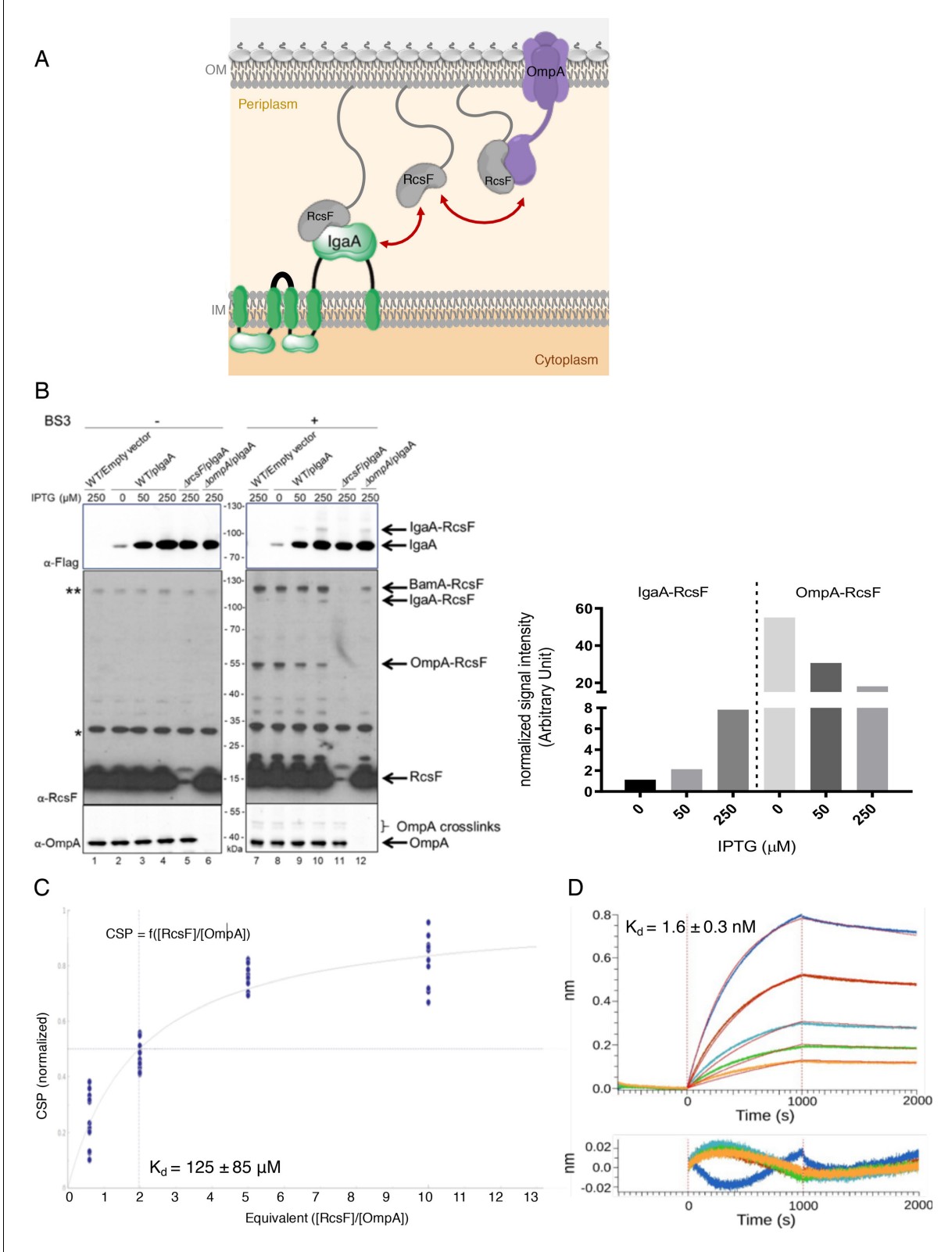

**Figure 5.** OmpA likely competes with the IM protein IgaA for RcsF. (**A**) Cartoon of the *E. coli* cell envelope, with IgaA in the IM and RcsF and OmpA in the OM. RcsF, an OM lipoprotein, interacts with OmpA. When in complex with RcsF, OmpA adopts its two-domain conformation, with an N-terminal β-barrel embedded in the OM and a C-terminal domain soluble in the periplasm. The periplasmic domain of OmpA interacts with RcsF, likely competing with IgaA for RcsF binding. Copy numbers are from *Li et al., 2014*. (**B**) Impact of over-expressing IgaA on the OmpA-RcsF complex (left). Cells

*Figure 5 continued on next page*

*Figure 5 continued*

harboring pSC231, an empty vector used as control, or pIgaA (pSC237, expressing IgaA-Flag$_3$ from an IPTG-inducible promoter) were harvested at mid-log phase and incubated without (lanes 1–6) or with (lanes 7–12) BS3. Protein samples were immunoblotted with α-Flag (upper panel), α-RcsF (middle panel), or α-OmpA$_{171-325}$ (lower panel) antibodies. IgaA-Flag$_3$ was expressed in WT (lanes 2–4 and 8–10), Δ*rcsF* (lanes 5 and 11), and Δ*ompA* cells (lanes 6 and 12) with the indicated IPTG concentrations. Quantitation (see *Figure 5—source data 1* for details) of the IgaA-RcsF and OmpA-RcsF complexes detected by the anti-RcsF antibodies is shown (right panel).* and **, non-specific bands detected by the polyclonal α-RcsF antibody. (C) Plot of the chemical shift perturbation (CSP) measured on OmpA resonances as a function of the RcsF:OmpA ratio. Only the residues with significant CSP (colored in blue in *Figure 4B*) are plotted and used to fit the $K_D$. (D) The interaction between RcsF and the periplasmic domain of IgaA was probed by biolayer interferometry (BLI). Sensortips carrying immobilized RcsF were dipped into increasing concentrations of IgaA (5.9, 8.9, 13.3, 20, 30 nM) from 0 to 1000s then into buffer (1000–2000s). Association and dissociation phases were fitted (red lines) to extract a $K_D$ value. Residuals from the fits are shown at the bottom of the panel (see *Figure 5—source data 2* for statistics).

The online version of this article includes the following source data and figure supplement(s) for figure 5:

**Source data 1.** Raw source data for *Figure 5B*.
**Source data 2.** Raw source data for *Figure 5D*.
**Figure supplement 1.** Estimation of the concentrations of IgaA and OmpA in the cell envelope.

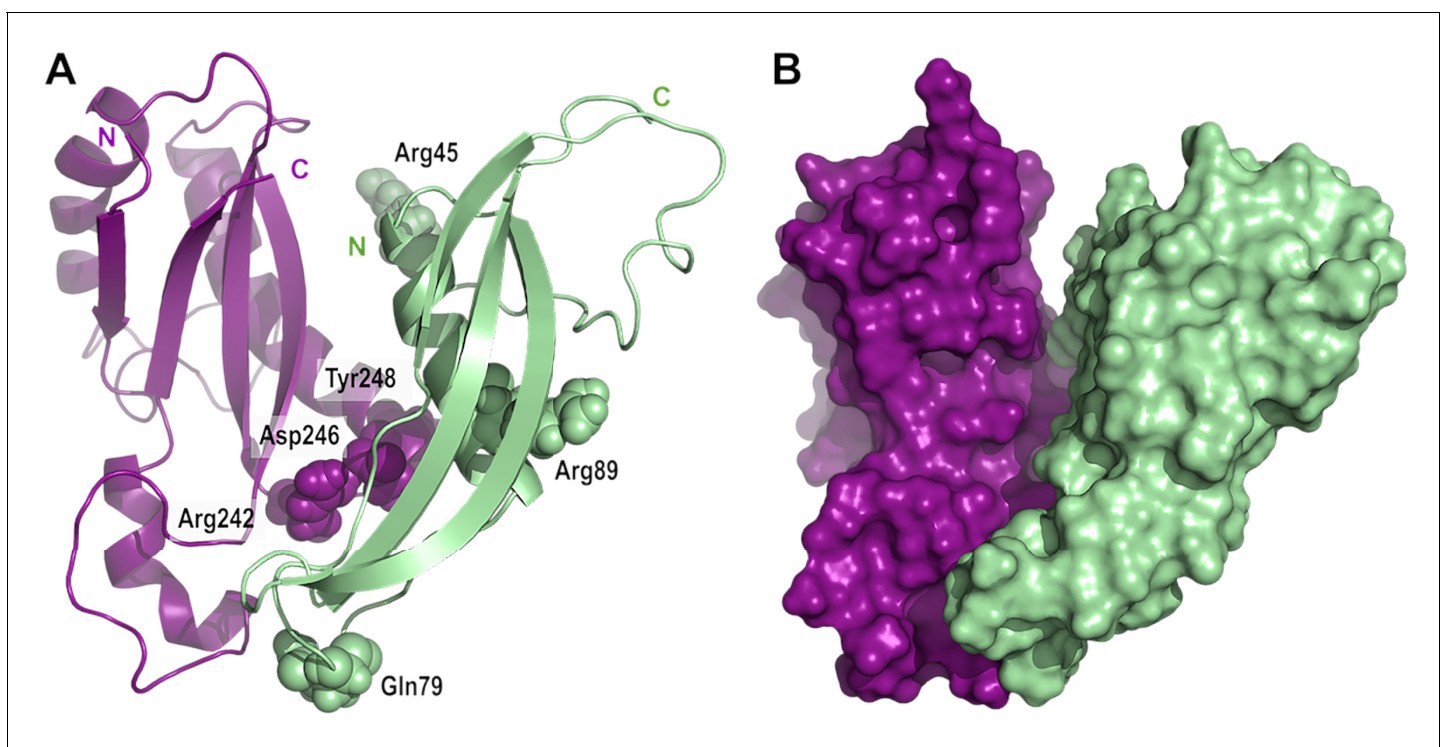

**Figure 6.** Model of the complex between RcsF and the C-terminal domain of OmpA. (A) Cartoon and (B) surface representation of a complex between RcsF (residues 44–134, colored in light green) and OmpA (residues 190–315, colored in purple). The atoms from the six residues mutated to DiZPK (RcsF$_{R45X}$, RcsF$_{Q79X}$, RcsF$_{R89X}$, OmpA$_{R242X}$, OmpA$_{D246X}$, and OmpA$_{Y248X}$) are represented as spheres with the van der Waals radius. This model, which shows a good surface complementarity between the two proteins, was obtained by protein-protein docking with constraints from the crosslinking experiments and further refined by all-atom molecular dynamics (50 ns). The N-terminal residue of RcsF points out in the same direction as the extremities of the periplasmic domain of OmpA, that is toward the OM, and this binding mode is compatible with the interaction of the periplasmic domain of OmpA with the peptidoglycan.

The online version of this article includes the following source data and figure supplement(s) for figure 6:

**Figure supplement 1.** The Rcs system responds to polymyxin exposure in cells lacking OmpA.
**Figure supplement 1—source data 1.** Raw source data for *Figure 6—figure supplement 1*.
**Figure supplement 2.** OmpA functions as a buffer for RcsF.

## OmpA functions as a buffer for RcsF

Finally, if OmpA does not allow RcsF to reach the surface, what could be the function of the OmpA-RcsF complex? On the basis of our results, we propose that the role of OmpA in Rcs is to modulate the activity of this system. Consider the equilibrium dissociation constants of the IgaA-RcsF and OmpA-RcsF complexes:

$$K_d^{\text{OmpA}-\text{RcsF}} = \frac{[\text{OmpA}][\text{RcsF}]}{[\text{OmpA}-\text{RcsF}]} \tag{1}$$

$$K_d^{\text{IgaA}-\text{RcsF}} = \frac{[\text{IgaA}][\text{RcsF}]}{[\text{IgaA}-\text{RcsF}]} \tag{2}$$

We estimate that OmpA's C-terminal domain is present in the periplasm at ~1 mM, RcsF at ~15 μM, and IgaA at ~1 μM (Materials and methods). If we take into account these estimates, then the following equation can be derived from *Equations 1 and 2* (Materials and methods):

$$[\text{IgaA}-\text{RcsF}] = \frac{K_d^{\text{ratio}}[\text{OmpA}-\text{RcsF}]}{1000 + K_d^{\text{ratio}}[\text{OmpA}-\text{RcsF}]} \tag{3}$$

where $K_d^{\text{ratio}} = \frac{K_d^{\text{OmpA}-\text{RcsF}}}{K_d^{\text{IgaA}-\text{RcsF}}}$.

Because OmpA is highly abundant, [OmpA] can be considered constant. Thus, according to *Equation 1*, the concentration of the OmpA-RcsF complex increases linearly with the concentration of RcsF in the periplasm (*Figure 6—figure supplement 2*). From *Equation 3*, we conclude that the concentration of IgaA-RcsF will also increase as a function of [RcsF] in the periplasm. However, in this case, the increase is not linear, but is decelerated and follows a hyperbolic curve (*Figure 6—figure supplement 2*), indicating that OmpA functions as a buffer for RcsF, negatively impacting its ability to activate Rcs. The buffering function of OmpA is nicely illustrated by the fact that exposure to low concentrations of polymyxin B causes higher Rcs induction in cells lacking *ompA* than in the wild type (*Figure 6—figure supplement 1*).

We acknowledge that the ~80,000 fold difference in affinity for RcsF that we measured between IgaA and OmpA (*Figure 5*) does not seem to support the function of OmpA as a buffer for RcsF: with such a high affinity for RcsF, IgaA should always outcompete OmpA and formation of the IgaA-RcsF complex should be constitutive, which is not the case (*Hussein et al., 2018*). To explain the apparent discrepancy between the in vivo observations and the in vitro measurements, we propose that intracellular factors such as attachment of the periplasmic domains of these proteins to their respective membrane anchors (we used the soluble periplasmic domains of OmpA, IgaA and RcsF for the in vitro measurements) and binding of OmpA to the peptidoglycan (*Reusch, 2012*) modulate the affinity of OmpA and IgaA for RcsF in vivo, allowing them to actually compete for RcsF. It is remarkable that OmpA, a protein that had mostly been known for its role in stabilizing the OM, is also involved in the network controlling the activity of Rcs, one of the most complex signal transduction systems in bacteria (*Cho et al., 2014*; *Hussein et al., 2018*; *Laloux and Collet, 2017*), whose correct functioning is critical for success of commensal Enterobacteriaceae and virulence of pathogens.

## Materials and methods

### Bacterial strains, primers, and plasmids

The bacterial strains and primers used in this study are listed in *Supplementary files 1* and *2*, respectively. The parental *E. coli* strain DH300 is a MG1655 derivative containing a deletion of the *lac* region; it also carries a chromosomal *rprA::lacZ* fusion at the λ phage attachment site to monitor Rcs activation (*Majdalani et al., 2002*). The *ompA*, *ompR* and *rcsF* deletion mutants were obtained by transferring the corresponding alleles from the Keio collection (kan[R]) (*Baba et al., 2006*) into DH300 (*Majdalani et al., 2002*) via P1 phage transduction, which was verified via PCR. To excise the kanamycin-resistance cassette, we used pCP20 as previously described (*Datsenko and Wanner, 2000*). To insert *ompA-His* (encoding OmpA with six histidines at the C-terminus), *ompA_{TH189}-His*,

$ompA_{TH243}$-His, and $ompX$-$ompA_{171-325}$ on the chromosome at the $ompA$ locus, we performed λ-Red recombineering (*Yu et al., 2000*) with the pSIM5-Tet plasmid (*Koskiniemi et al., 2011*). First, a $cat$-$sacB$ cassette encoding chloramphenicol acetyl transferase (cat) and SacB, a protein conferring sensitivity to sucrose, was amplified from strain CH1990 using primers 'ompA_delCmSB F' and 'ompA_delCmSB R'. The resulting PCR product shared 40 bp of homology to the 5' UTR and 3' UTR of $ompA$ at its 5' and 3' ends, respectively, and was used for λ-Red recombineering (*Yu et al., 2000*). We selected transformants for chloramphenicol resistance and verified that the $cat$-$sacB$ cassette replaced $ompA$ by sequencing across the junctions. The $cat$-$sacB$ cassette was subsequently replaced by one of the versions of $ompA$ ($ompA$-His, $ompA_{TH189}$-His, $ompA_{TH243}$-His, or $ompX$-$ompA_{171-325}$) using λ-Red recombineering and negative selection on sucrose-containing medium (*Thomason et al., 2014*; *Gennaris et al., 2015*). $ompA$-His, $ompA_{TH189}$-His, and $ompA_{TH243}$-His were amplified *via* PCR using primers 'cmSB to OmpA_F' and 'cmSB to OmpAhis_R' and plasmids pPR11, pPR11$_{189thrombin}$, and pPR11$_{243thrombin}$ (see below) as templates, respectively. The PCR product of $ompX$-$OmpA_{171-325}$ was generated using primers 'JLEo20-F-Chrom-OmpX' and 'JLEo22-R-Chrom-OmpACt', and pJLE17-OmpX-OmpACter as template. Strains were verified through DNA sequencing. To delete $ompA_{171-325}$, we performed λ-Red recombineering on the chromosome at the $ompA$ locus in a way similar to the description above. Briefly, the kanamycin cassette from the strain SEN588 (Δ$ompA$::$kan$) was PCR-amplified using primers 'ompAc_delKm F' and 'ompAc_delKm R' to encompass the flanking regions of $ompA_{171-325}$. We selected transformants for kanamycin resistance.

The plasmids used in this study are described in *Supplementary file 3*. pSC253, encoding RcsF, was constructed *via* digestion of pSC202 (*Cho et al., 2014*) with $Kpn$I and $Hind$III and insertion of the generated fragment into pBAD18 (*Guzman et al., 1995*). To generate the $rcsF$ variants containing an amber codon (TAG) at selected positions, site-directed mutagenesis was performed on pSC253 using primers described in *Supplementary file 2*. To generate pPR11, $ompA$-His was PCR-amplified using primers 'OmpA(NcoI) F' and 'OmpA-his(XbaI) R' and chromosomal *E. coli* DNA as template. The PCR product was inserted into pDSW204 restricted with $Nco$I and $Xba$I, generating pPR11. To insert thrombin cleavage sites after residues 189 and 243 in OmpA, site-directed mutagenesis was performed using primer pair '189VPRGS thr_F' and '189VPRGS thr_R' and primer pair '243LVPR thr_F' and '243LVPR thr_R', respectively, on pPR11, yielding pPR11$_{189thrombin}$ [OmpA(189-Val-*Val-Pro-Arg-Gly-Ser*-Gln-190)] and pPR11$_{243thrombin}$ [OmpA(243-Ile-*Leu-Val-Pro-Arg*-Gly-244)], respectively. pJLE17-OmpX-$OmpA_{171-325}$ (encoding OmpA$_X$) was constructed as follows. The two PCR products corresponding to $ompX$ and $OmpA_{171-325}$ were generated using primer pair 'OmpX (NcoI)F' and 'OmpX-OmpAc_R' and primer pair 'OmpX-OmpAc_F' 'JLEo16-R-XbaI-OmpACt', respectively, and chromosomal *E. coli* DNA as template. To join the two PCR fragments, overlapping PCR was performed using primers 'OmpX(NcoI)F' and 'JLEo16-R-XbaI-OmpACt', generating the PCR product encoding OmpA$_X$. Next, the PCR product of $OmpA_X$ was digested with $Nco$I and $Xba$I and ligated with pDSW204 pre-digested with $Nco$I and $Xba$I, yielding pJLE17-OmpX-OmpACter. pKiD22, expressing OmpA$_{Pal}$, was constructed as follows. The sequence encoding the signal sequence and lipobox of the lipoprotein Pal was PCR-amplified using primers 'Palss(NcoI)F' and 'Palss-OmpAc_R' and the sequence encoding $OmpA_{171-325}$ using primers 'Palss-OmpAc_F' and 'OmpA_stop_Flag$_3$(KpnI)R'. Chromosomal *E. coli* DNA was used as template. To join the two PCR fragments, overlapping PCR was performed using primers 'Palss(NcoI)F' and 'OmpA_stop_-Flag$_3$(KpnI)R'. The PCR product encoding ssPal-OmpA$_{171-325}$ was digested with $Nco$I and $Kpn$I and ligated with pDSW204, pre-digested with $Nco$I and $Kpn$I, yielding pKiD22.

To generate pPR21, site-directed mutagenesis was performed on pPR11 using primers 'OA_stopTGA_F' and 'OA_stopTGA_R' to insert a stop codon upstream of the 6xHis tag. To generate the $ompA$ variants containing an amber codon (TAG) at selected positions, site directed mutagenesis was performed on pPR21 using primers described in *Supplementary file 2*. To obtain pKiD5, OmpA$_{186-325}$ with an N-terminal Strep-tag but no signal sequence was PCR-amplified using primers 'KiDo14-F-NdeI-Strep-OmpACTD' and 'KiDo15-R-SacI-Rbs-OmpACTD' and chromosomal *E. coli* DNA as template. The PCR product was digested with $Nde$I and $Sac$I and inserted into pET21a. To prepare a version of pAM238 containing $lacI^q$, a $trc$ promoter, and a triple Flag tag (Flag$_3$), we prepared a PCR product using primers 'lacIq NsiI_F' and 'flag3-KpnI_R' and pMER77 as template (*Hemmis et al., 2011*). This product was digested with $Nsi$I and $Kpn$I and ligated with pAM238 pre-digested with $Pst$I and $Kpn$I. To reduce the basal activity of the $trc$ promoter, we modified the −10

region (from TATAAT to CATTAT) and the −35 region (from TTTACA to TTGACA), generating pSC231 (*Weiss et al., 1999*). The coding sequences for OmpA and IgaA were obtained by PCR-amplification using primer pair 'OmpA(NcoI) F' and 'OmpA XbaI flag R' and primer pair 'IgaA(NcoI) F' and 'IgaA XbaI flag3 R', respectively. Each product was digested with *NcoI* and *XbaI* and ligated into pSC231 pre-digested with the same enzymes, generating pPR4 and pSC237, respectively.

## In vivo site-specific photo-crosslinking using DiZPK

We used the site-specific photo-crosslinking method described previously (*Cho et al., 2014*) with some modifications. To incorporate $N^6$-((3-(3-methyl-3$H$-diazirin-3-yl)propyl)carbamoyl)-$L$-lysine (DiZPK) into RcsF, we used the pSup-Mb-DIZPK-RS plasmid encoding an evolved *Methanosarcina barkeri* pyrrolysyl-tRNA synthetase (PylRS) and an optimized $\mathrm{tRNA}_{\mathrm{CUA}}^{\mathrm{Pyl}}$ suppressor (*Zhang et al., 2011*). DH300 Δ*rcsF* (PL358) cells were co-transformed with pSup-Mb-DIZPK-RS and one of the plasmids containing an amber codon in *rcsF* in pSC253. Cells were grown in 3-(N-morpholino)propane-sulfonic acid (MOPS) minimal medium supplemented with 0.2% glucose, 0.2% L-arabinose (MOPS-glucose/arabinose minimal medium), and 0.8 mM DiZPK (no supplement of other amino acids; see the reasons for using MOPS medium below) (Neidhardt, Bloch, and Smith 1974). Cell cultures were grown to an $OD_{600}$ of 1-1.2 and 1 mL of samples was irradiated with UV light at 365 nm or left unirradiated for 10 min. Cells were precipitated with trichloroacetic acid (TCA), and the pellets were washed with acetone and solubilized in 60 µL of SDS-sample buffer before further analysis. We used a similar method to incorporate DiZPK into OmpA with minor modifications; DH300 Δ*ompA* (PR46) cells were co-transformed with pSup-Mb-DIZPK-RS and one of the plasmids containing an amber codon in *ompA* in pPR21. Cells were grown in LB medium supplemented with 0.2% L-arabinose, 200 µM IPTG and 1 mM DiZPK. Cell cultures were grown to an $OD_{600}$ of 1 and 0.75 mL of samples was irradiated with UV light at 365 nm or left unirradiated for 10 min. Cells were precipitated with TCA, and the pellets were washed with acetone and solubilized in 100 µL of SDS-sample buffer before further analysis.

In previous experiments incorporating *p*-benzoyl-L-phenylalanine into RcsF, we used LB as the growth medium (*Cho et al., 2014*). However, we found that the expression levels of the DiZPK-containing RcsF mutant proteins were substantially lower when cells were grown in LB (data not shown); in contrast, the expression levels of RcsF were greatly enhanced in MOPS-glucose/arabinose minimal medium (data not shown). Therefore, we used MOPS-glucose/arabinose minimal medium for all photo-crosslinking experiments involving DiZPK-containing RcsF variants.

## Synthesis of DiZPK

DiZPK was synthesized as described previously (*Zhang et al., 2011*).

## Immunoprecipitation of RcsF-containing complexes and thrombin cleavage

Protein samples in SDS-sample buffer (without reducing agent) were denatured for 15 min at 65°C and 15 min at 95°C with vigorous shaking. Next, the samples were diluted in 750 µL of KI buffer (50 mM Tris-HCl [pH 8], 2% Triton X-100, 150 mM NaCl, 1 mM EDTA) and centrifuged at 4°C for 3 min at 12,000 x *g* to harvest the PG. Photo-crosslinked RcsF complexes were immunoprecipitated by adding 1 µL of undiluted guinea pig anti-RcsF antibody (*Cho et al., 2014*) and 10 µL of protein A/G magnetic beads (Pierce); samples were incubated for 1 hr on a wheel at room temperature. After three washes with 500 µL of KI buffer, RcsF complexes were eluted with 20 µL of glycine buffer (100 mM glycine [pH 1.5], 0.1% Triton X-100) after 10 min of incubation at room temperature. Proteins samples were neutralized with 2 µL of 1.5 M Tris-HCl [pH 8.8] and diluted with 18 µL of KI buffer before SDS-PAGE and immunoblotting or thrombin cleavage. For thrombin cleavage, 20 µL of the elution samples were incubated for 1 hr at room temperature with 1 µL of thrombin (thrombin from bovine plasma, Sigma). Samples were analyzed *via* SDS-PAGE and immunoblotting as indicated in the figure legends.

## Immunoblotting and antibodies

Protein bands were transferred from the gels onto nitrocellulose membranes (Millipore) using a semi-dry electroblotting system. The membranes were blocked with 5% skim milk. The rest of the

immunoblot steps were performed using standard protocols. Signal from antibody binding was visualized by detecting chemiluminescence from the reaction of horseradish peroxidase with luminol. Polyclonal RcsF antibodies were purified against the carboxy-terminal domain of RcsF as previously described (*Cho et al., 2014*) and used at a dilution of 1:20,000 in 1% skim milk in 50 mM Tris-HCl [pH 7.6], 0.15 M NaCl and 0.1% Tween 20 (TBST). Since we found that the anti-OmpA antibody only recognizes the periplasmic domain (data not shown), we used an antibody directed against loop 4 of the ß-barrel of OmpA to detect OmpA$_{1-170}$. The anti-OmpA antibodies are gifts from the Lloubes and Bernstein laboratories (*Cascales et al., 2002*; *Hussain and Bernstein, 2018*). These antibodies were used at dilutions of 1:20,000 and 1:10,000, respectively. The anti-His antibody conjugated to horseradish peroxidase (Qiagen) was used at a dilution of 1:5000. The anti-Flag M2 monoclonal antibody (F1804, Sigma) was used at a dilution of 1:20,000.

### In vivo 3,3'-dithio-bis[sulfosuccinimidylpropionate] and bis (sulfosuccinimidyl)-suberate (BS3) crosslinking

DTSSP and BS3 (CovaChem) are bifunctional primary amine crosslinkers; DTSSP contains a disulfide bond in its spacer arm. In vivo crosslinking has been performed as described previously (*Cho et al., 2014*). The media used are MOPS-glucose minimal medium (*Neidhardt et al., 1974*; *Figure 3*) and LB (*Figure 5*).

### β-Galactosidase assay

*E. coli* cells were grown in MOPS-glucose minimal medium (*Neidhardt et al., 1974*) or LB to mid-log phase (OD$_{600}$ = 0.4–0.6). β-galactosidase activity was measured as described previously (*Zhang and Bremer, 1995*).

### Expression and purification of RcsF, OmpA$_{186-325}$ and of the periplasmic domain of IgaA

Expression and purification of RcsF with a C-terminal His-tag were performed as previously described (*Leverrier et al., 2011*). *E. coli* BL21(DE3) cells harboring pKiD5 expressing N-terminal Strep-tagged OmpA$_{186-325}$ were grown at 37°C in M9-glucose minimal medium containing 1 g/L $^{15}$NH$_4$Cl (99%, $^{15}$N; Eurisotop) to uniformly label the protein with $^{15}$N. The expression was induced by adding 1 mM IPTG at an OD$_{600}$ of 0.5. After a 5 hr induction, cells were harvested by centrifugation and resuspended in 20 mL of lysis buffer (20 mM Tris-HCl, 500 mM NaCl [pH8], containing a protease inhibitor cocktail (Complete, Roche)). Re-suspended cells were stored at −20°C. Frozen cells were thawed on ice and lysed by two passages through a French pressure cell at 1500 psi. The soluble fraction was isolated after centrifugation for 1 hr at 40,000 x *g* at 4°C. The supernatant was filtered through 0.45 µm filters and loaded onto a 5 mL Strep-Tactin column (IBA, Lifesciences), previously equilibrated in buffer A (50 mM NaH$_2$PO$_4$, 100 mM NaCl [pH7.0]). After a washing step with buffer A, elution was performed with buffer A supplemented with 2.5 mM D-desthiobiotin. The sample was then further purified using size-exclusion chromatography on a HiLoad 16/60 Superdex 75 column (GE Healthcare) using buffer A. To express IgaA$_{361-654}$ with a C-terminal His-tag, *E. coli* SHuffle T7 cells harboring pSC211 were grown at 37°C in LB. Expression of the protein was induced by adding 1 mM IPTG at an OD$_{600}$ of 0.5. After a 5 hr induction, cells were harvested by centrifugation and resuspended in 20 mL of lysis buffer. Re-suspended cells were stored at −20°C. Frozen cells were thawed on ice and processed as explained above for Strep-tagged OmpA$_{186-325}$. IgaA$_{361-654}$ was purified using Ni-NTA agarose beads (5 mL; IBA Lifescience), previously equilibrated in buffer B (20 mM NaH$_2$PO$_4$, 150 mM NaCl [pH7.5]). After washing the resin with buffer B supplemented with 20 mM imidazole, proteins were eluted with five column volumes of buffer B supplemented with 200 mM imidazole. As a final purification step, a size-exclusion chromatography was performed using a HiLoad 16/60 Superdex 200 column (GE Healthcare) with buffer B. Purity was checked *via* SDS-PAGE with Coomassie Staining and concentration was performed using Vivaspin Turbo apparatus (Sartorius) with a 5 kDa molecular weight cut-off.

### NMR titration experiments of $^{15}$N-labelled OmpA$_{186-325}$ with RcsF

50 µM $^{15}$N-labelled OmpA$_{186-325}$ in buffer A at 25°C was titrated with increasing concentrations of unlabelled RcsF in buffer A to reach 2 and 10 RcsF/OmpA$_{186-325}$ molar ratios. To follow $^{15}$N-$^{1}$H

resonances chemical shifts perturbations, $^{15}$N-$^{1}$H BEST-TROSY-HSQC correlation experiments were recorded at 25°C using Bruker AVANCE spectrometer operating at 700 MHz proton frequency equipped with a TCI cryoprobe (*Favier and Brutscher, 2011*). OmpA$_{186-325}$ assignments were transposed from BMRB entry 25030. Chemical shift perturbations (CSP) were calculated on a per-residue basis for the highest substrate-to-protein ratio as described previously (*Egan et al., 2018*). Spectra were processed with Topspin 3.57 (Bruker) and analyzed with ccpnmr 3 (https://www.ccpn.ac.uk).

## Western blot analysis

Quantitation of the intensity of the bands was done by optical densitometry and analyzed using ImageQuant TL processing software (ImageQuant TL v8.1.0.0, GE HEalthcare). After background correction, the values were normalized by the intensity of non-specific bands detected by the polyclonal antibody.

## Biolayer interferometry

Biolayer Interferometry Experiments (BLI) were recorded on an OctetRED96e (Fortebio) using streptavidin (SA) biosensors (Fortebio). To biotinylate RcsF, RcsF-His (5 mg/mL in 0.1 M MES [pH 5.5]) was incubated with Biotin-LC-Hydrazide (1.25 mM, final concentration; Sigma) and N-(3-Dimethylaminopropyl)-N'-ethylcarbodiimide (EDC; 6.5 mM, final concentration) during 2 hr at 22°C under agitation. Biotininylated RcsF was dialysed against buffer C (10 mM Hepes, 150 mM NaCl [pH7.5]) and immobilized at 5 µg/mL onto SA tips in buffer C supplemented with 0.02% Tween-20 to reach ~3.5 nM of immobilization level. RcsF-loaded biosensors were dipped into different concentrations of the periplasmic domain of IgaA (5.9 nM to 30 nM) in buffer C at 23°C. Kinetics were recorded with 1000s association and 1000s dissociation phases, and repeated four times with 10 mM HCl pulses (18 s in total) used for regeneration between cycles. Sensorgrams were subtracted for contribution of buffer alone and binding of non-functionalized biosensors. Kinetic analysis of the data was performed using 1:1 interaction model in the ForteBio data analysis software. $K_D$ obtained from four independent injection series were averaged and produced a $K_D$ of 1.6 nM with a standard deviation of 0.3.

## Estimation of the expression levels of IgaA

Overexpressed IgaA has a triple Flag tag (IgaA-Flag$_3$). To compare the expression levels of IgaA to those of OmpA, we generated a triple Flag-tagged version of OmpA (OmpA-Flag$_3$); this variant was expressed from an IPTG-inducible plasmid in the *ompA* strain. If we compare the intensity of the signal corresponding to OmpA-Flag$_3$ detected either by the anti-OmpA or the anti-Flag antibodies, we estimate the anti-Flag antibodies to be ~25 times more sensitive than the anti-OmpA antibodies (lanes 1–4 in *Figure 5—figure supplement 1*). With this value in hand, we can now compare the expression levels of IgaA-Flag$_3$ to those of untagged OmpA produced from the chromosome and detected with the anti-OmpA antibodies (lanes 5–9 in *Figure 5—figure supplement 1*); because the intensities of OmpA and IgaA-Flag$_3$ in lane eight are similar, we estimate that IgaA-Flag$_3$ is 25 times less abundant than OmpA when IgaA is expressed with 250 µM IPTG; likewise, we estimate the levels of IgaA-Flag$_3$ to be ~1/125 (0.8%) of those of OmpA when IgaA is expressed with 50 µM IPTG. Because the concentration of OmpA is ~1 mM, we estimate the concentration of overexpressed IgaA to be 40 and 8 µM, respectively.

## Molecular modelling

HADDOCK 2.4 web server (*van Zundert et al., 2016*; *Wassenaar et al., 2012*) was used for protein-protein docking using the structures 2MQE for OmpA-CTD (*Ishida et al., 2014*) and 2L8Y for RcsF (*Rogov et al., 2011*). Six residues identified from the crosslinking experiments were considered as active during the docking calculations: 242, 246, and 248 for OmpA and 45, 79, and 89 for RcsF. The resulting clusters were inspected visually, and the one compatible with the interaction between the periplasmic domain of OmpA and the peptidoglycan (*Park et al., 2012*) was selected for further refinement using molecular dynamics. Molecular dynamics simulations were carried out with GROMACS version 2020.1 (*Abraham et al., 2015*) using the OPLS-AA (*Jorgensen et al., 1996*) force field. Each system was energy-minimized until convergence using a steepest descents algorithm. Molecular dynamics with position restraints was then performed (50 ps NVT and 50 ps NPT),

followed by the production run of 50 ns. During the position restraints and production runs, the V-rescale and Parrinello-Rahman methods were used for temperature and pressure coupling, respectively. Electrostatics were calculated with the particle mesh Ewald method. The P-LINCS algorithm was used to constrain bond lengths, and a time step of 2 fs was used throughout.

## Estimation of the concentrations of OmpA, RcsF and IgaA in the periplasm

According to *Li et al., 2014*, each cell contains ~200,000 molecules of OmpA in rich media, which corresponds to ~$3.448 \times 10^{-19}$ mol (200,000/Avogadro constant). If we consider the volume of an *E. coli* cell to be $10^{-18}$ m$^3$ ($10^{-15}$ L) and compare it to a cube with 1 μm edges (*E. coli* has 0.5 μm in width and 2 μm in length; EcoliWiki ecoliwiki.net/colipedia/index.php/Escherichia_coli), we calculate that the cellular concentration of OmpA is ~$3.448 \times 10^{-4}$ M. The volume of the envelope being ~30% of the total cell volume (*Stock et al., 1977*), we calculate that the concentration of the C-terminal domain of OmpA in the periplasm is ~1 mM. If we perform the same calculations for RcsF and IgaA (~3000 and ~200 copies/cell, respectively), we find that their periplasmic concentrations are ~15 μM and ~1 μM, respectively.

## OmpA functions as a buffer for RcsF

The equilibrium dissociation constants for the OmpA-RcsF and IgaA-RcsF complexes are:

$$K_d^{\mathrm{OmpA-RcsF}} = \frac{[\mathrm{OmpA}][\mathrm{RcsF}]}{[\mathrm{OmpA-RcsF}]} \tag{4}$$

$$K_d^{\mathrm{IgaA-RcsF}} = \frac{[\mathrm{IgaA}][\mathrm{RcsF}]}{[\mathrm{IgaA-RcsF}]} \tag{5}$$

If we divide *Equation 4* with *Equation 5*, we obtain:

$$\frac{K_d^{\mathrm{OmpA-RcsF}}}{K_d^{\mathrm{IgaA-RcsF}}} = \frac{[\mathrm{OmpA}][\mathrm{IgaA-RcsF}]}{[\mathrm{IgaA}][\mathrm{OmpA-RcsF}]} \tag{6}$$

If we replace $\frac{K_d^{\mathrm{OmpA-RcsF}}}{K_d^{\mathrm{IgaA-RcsF}}}$ by $K_d^{\mathrm{ratio}}$, then we can rearrange *Equation 6* to,

$$\frac{[\mathrm{IgaA-RcsF}]}{[\mathrm{IgaA}]} = \frac{K_d^{\mathrm{ratio}}[\mathrm{OmpA-RcsF}]}{[\mathrm{OmpA}]} \tag{7}$$

Under physiological conditions, we estimate the concentration of IgaA to be ~1 μM (see above). Thus, because [IgaA] + [IgaA-RcsF]=1 μM, *Equation 7* can be successively rearranged to,

$$\frac{[\mathrm{IgaA-RcsF}]}{1-[\mathrm{IgaA-RcsF}]} = \frac{K_d^{\mathrm{ratio}}[\mathrm{OmpA-RcsF}]}{[\mathrm{OmpA}]} \tag{8}$$

$$\frac{1-[\mathrm{IgaA-RcsF}]}{[\mathrm{IgaA-RcsF}]} = \frac{[\mathrm{OmpA}]}{K_d^{\mathrm{ratio}}[\mathrm{OmpA-RcsF}]} \tag{9}$$

$$\frac{1}{[\mathrm{IgaA-RcsF}]} = \frac{[\mathrm{OmpA}]}{K_d^{\mathrm{ratio}}[\mathrm{OmpA-RcsF}]} + 1 \tag{10}$$

$$\frac{1}{[\mathrm{IgaA-RcsF}]} = \frac{[\mathrm{OmpA}] + K_d^{\mathrm{ratio}}[\mathrm{OmpA-RcsF}]}{K_d^{\mathrm{ratio}}[\mathrm{OmpA-RcsF}]} \tag{11}$$

yielding:

$$[\mathrm{IgaA-RcsF}] = \frac{K_d^{\mathrm{ratio}}[\mathrm{OmpA-RcsF}]}{[\mathrm{OmpA}] + K_d^{\mathrm{ratio}}[\mathrm{OmpA-RcsF}]} \tag{12}$$

In *Equation 4*, [OmpA], which is ~1000 µM, can be considered as a constant. Therefore, *Equations 4 and 14* become:

$$[\text{OmpA} - \text{RcsF}] = \frac{1000\,[\text{RcsF}]}{K_d^{\text{OmpA}-\text{RcsF}}} \tag{13}$$

$$[\text{IgaA} - \text{RcsF}] = \frac{K_d^{\text{ratio}}[\text{OmpA} - \text{RcsF}]}{1000 + K_d^{\text{ratio}}[\text{OmpA} - \text{RcsF}]} = \frac{[\text{OmpA} - \text{RcsF}]}{1000/K_d^{\text{ratio}} + [\text{OmpA} - \text{RcsF}]}$$

$$= \frac{\frac{1000\,[\text{RcsF}]}{K_d^{\text{OmpA}-\text{RcsF}}}}{1000/K_d^{\text{ratio}} + \frac{1000\,[\text{RcsF}]}{K_d^{\text{OmpA}-\text{RcsF}}}} \tag{14}$$

Thus, from *Equation 14*, we conclude that whereas [OmpA-RcsF] increases linearly to [RcsF], [IgaA-RcsF] increases proportionally, but not linearly, to [RcsF]. Thus, OmpA functions as a buffer for RcsF.

### Analysis of protein structures

Protein structures were downloaded from the Protein Data Bank (http://www.rcsb.org; PDB codes are indicated) and visualized using PyMOL Molecular Graphics System (Version 2.3.4, Schrödinger, LLC). FASTA protein sequences were downloaded from Uniprot (http://www.uniprot.org/).

## Acknowledgements

We are indebted to Marie Renault (CNRS, Toulouse) for discussion and to Harris Bernstein (NIH, USA), Eric Cascales and Roland Lloubes (CNRS, Marseille) for sharing antibody with us. We are grateful to Alexandra Gennaris (UCLouvain), Pauline Leverrier (UCLouvain) and Camille Goemans (EMBL, Heidelberg) for critically reading the manuscript and providing feedback. We thank the members of the lab for helpful discussions, and A Boujtat for assistance in experiments. This work used the platforms of the Grenoble Instruct-ERIC center (ISBG; UMS 3518 CNRS-CEA-UGA-EMBL) within the Grenoble Partnership for Structural Biology (PSB), supported by FRISBI (ANR-10-INBS-05–02) and GRAL, financed within the University Grenoble Alpes graduate school (Ecoles Universitaires de Recherche) CBH-EUR-GS (ANR-17-EURE-0003). Authors acknowledge the SPR/BLI platform personal, Jean-Baptiste Reiser and Anne Chouquet, for their help and assistance. KD and RB are FRIA research fellows and JL is 'Chargée de Recherches' of the Fonds de la Recherche Scientifique FRS-FNRS. This work was supported by grants from FRFS-WELBIO, from the FRS-FNRS, and from the Fédération Wallonie-Bruxelles (ARC 17/22–087).

## Additional information

### Funding

| Funder | Grant reference number | Author |
| --- | --- | --- |
| Fonds pour la Formation à la Recherche dans l'Industrie et dans l'Agriculture | | Kilian Dekoninck<br>Robin Bevernaegie |
| Fonds De La Recherche Scientifique - FNRS | | Kilian Dekoninck<br>Juliette Létoquart<br>Robin Bevernaegie<br>Olivia Dehu<br>Benjamin Elias<br>Seung-Hyun Cho |
| FRFS-WELBIO | | Jean-Francois Collet |
| FRISBI | ANR-10-INBS-05-02 | Cédric Laguri<br>Jean-Pierre Simorre |
| GRAL | | Cédric Laguri |

| | | Jean-Pierre Simorre |
| --- | --- | --- |
| CBH-EUR-GS | ANR-17-EURE-0003 | Cédric Laguri
Jean-Pierre Simorre |
| Fédération Wallonie-Bruxelles | ARC 17/22-087 | Jean-Francois Collet |

The funders had no role in study design, data collection and interpretation, or the decision to submit the work for publication.

### Author contributions
Kilian Dekoninck, Formal analysis, Validation, Investigation, Writing - review and editing; Juliette Létoquart, Formal analysis, Validation, Investigation, Writing - original draft, Writing - review and editing; Cédric Laguri, Pascal Demange, Robin Bevernaegie, Jean-Pierre Simorre, Olivia Dehu, Investigation; Bogdan I Iorga, Software, Investigation; Benjamin Elias, Supervision, Investigation; Seung-Hyun Cho, Conceptualization, Formal analysis, Supervision, Validation, Investigation, Methodology, Writing - original draft, Writing - review and editing; Jean-Francois Collet, Conceptualization, Formal analysis, Supervision, Funding acquisition, Writing - original draft, Project administration, Writing - review and editing

### Author ORCIDs
Kilian Dekoninck (iD) https://orcid.org/0000-0002-5093-1343
Robin Bevernaegie (iD) http://orcid.org/0000-0003-1605-9253
Bogdan I Iorga (iD) http://orcid.org/0000-0003-0392-1350
Benjamin Elias (iD) http://orcid.org/0000-0001-5037-3313
Seung-Hyun Cho (iD) http://orcid.org/0000-0002-5548-4239
Jean-Francois Collet (iD) https://orcid.org/0000-0001-8069-7036

### Decision letter and Author response
Decision letter https://doi.org/10.7554/eLife.60861.sa1
Author response https://doi.org/10.7554/eLife.60861.sa2

## Additional files

### Supplementary files
- Supplementary file 1. Strains used in this study.
- Supplementary file 2. Primers used in this study.
- Supplementary file 3. Plasmids used in this study.
- Transparent reporting form

### Data availability
All data generated or analyzed during this study are included in the manuscript and supporting files. Source data files have been provided for Figures 3, 4A, 4B, 5B, 5D and 6 supplement 1.

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
