## [Decision Letter]

**Acceptance summary:**

The Rcs phosphorelay plays a critical role in enabling *E. coli* and related organisms to respond to outer membrane stress. Prior work has demonstrated that the outer membrane, β-barrel protein OmpA influeces Rcs activity via the protein RcsF, but the molecular details of this important interaction have remained incompletely defined. This paper presents evidence indicating that RcsF interacts with the periplasmic globular domain of OmpA rather than through binding of the N-terminus of RcsF to the β-barrel portion of OmpA, as previously suggested. Based on these and other findings, the authors propose a compelling model in which the RcsF-OmpA interaction in the periplasm helps to buffer the ability of RcsF to bind IgaA and induce the Rcs phoshporelay.

**Decision letter after peer review:**

[Editors’ note: the authors submitted for reconsideration following the decision after peer review. What follows is the decision letter after the first round of review.]

Thank you for submitting your work entitled "OmpA and IgaA compete for the lipoprotein stress sensor RcsF across the periplasmic space, regulating Rcs activity" for consideration by *eLife*. Your article has been reviewed by three peer reviewers, and the evaluation has been overseen by a Reviewing Editor and a Senior Editor. The reviewers have opted to remain anonymous.

Our decision has been reached after consultation between the reviewers. Based on these discussions and the individual reviews below, we regret to inform you that your work will not be considered further for publication in *eLife*.

The reviewers each appreciated the attention to an important and still incompletely understood problem, namely understanding how RcsF helps *E. coli* cells respond to cell envelope stress. The reviewers also felt that the results related to the RcsF-OmpA interaction were likely to be of some interest, particularly those studying and following this sub-field. However, as one reviewer notes, the work here helps to rule out some possibilities for how RcsF works, but it did not fully support the competition model proposed. Additionally, there were concerns about how the biochemical findings reported here impact our understanding of Rcs signaling in vivo. The full reviews are included below and are quite constructive so we hope they will be of use to you moving forward.

Reviewer #1:

In this manuscript the authors find a new function for OmpA in the regulation of the RcsFBC system through a direct interaction with the periplasmic domain of the OmpA protein. Through biochemical and genetic experiments they elegantly establish that the carboxy-terminal domain of OmpA binds and suppresses the activation of RcsF that occurs when OmpA is deleted. The data is of high quality and the work resolves some controversies in the field and should be of general interest.

Specific comments:

The NMR results confirm that conformational changes in these residues occur when RcsF binds to OmpA, not that these residues are those that are bound directly, and though it is a distinct possibility that these residues form a binding interface the authors should be more cautious in the interpretation of their results.

The authors should be cautious about comparing the affinity of the interaction between IgaA and RcsF versus that between OmpA and RcsF based on the results with copy number, as many other factors include protein folding and accessary proteins/peptidoglycan binding could play a role in vivo.

The authors acutely point out the fact that it is hard to separate the effects of Bam binding to RcsF from its role in folding OmpA. However, does a periplasmic expressed carboxyl-terminal domain of OmpA (made as a fusion with a cleaved signal sequence) suppress the activation of RcsF in a OmpA null mutant, and if it does then a Bam null mutation could be added and would likely determine no effect on suppression. This experiment would rule out the rare possibility that the OmpX fusion has some unexpected binding through the OmpX domain.

Reviewer #2:

The outer-membrane linked lipoprotein RcsF is necessary for many signals that turn on the Rcs phosphorelay. The current model for signaling suggests that RcsF, when "activated" by an OM or PG stress, interacts with IgaA, relieving IgaA repression of Rcs signaling. How activation occurs, and in what way RcsF might be sequestered in the absence of signal and released upon signalling is the focus of the work here and would be of general interest in understanding this complex phosphorelay and its unique signaling mechanism. Previous findings had shown that parts of RcsF were exposed on the cell surface, and that RcsF interacted with outer membrane proteins, including OmpA, leading to a suggestion that RcsF might be threaded through the center of some porins (not necessarily but possibly OmpA) to the cell surface. This paper uses cross-linking and some chimeric protein constructs to suggest that RcsF binds to the periplasmic, rather than the β-barrel domain of OmpA, and that this binding acts to sequester RcsF. There are some major concerns that compromise the impact of this work: Many of the experiments require additional controls to confirm that they are assaying what they are said to assay, and the relationship of these findings to the critical in vivo signaling pathway is not currently well supported. Overall, it is not clear that the new information provided here adds significantly to our understanding of how RcsF signals; it may rule out some possibilities.

1) Figure 1: Site-specific crosslinking of OmpA and RcsF: RcsF is expressed from a high copy plasmid, and OmpA is very abundant. What is the evidence for site-specific cross-linking? At a minimum, does wild-type RcsF without the DiZPK fail to bring down any OmpA when subjected to UV and immunoprecipitation. It would seem a comparison of amounts in such a control would help to support that the cross-linking is due to the cross-linker (and is thus site specific).

2) In the Figure 1—figure supplement 1: Some of these are said to be positive (Q28, Q121 (mislisted in text as N121) but show very low amounts in upper panel (with RcsF), and some at bottom have very low OmpA signals. Why are these not in the same ratios? Shouldn't they be the same complex?

3) Figure 3D: This in vivo experiment is a critical part of the argument that only the periplasmic region of OmpA is critical for keeping RcsF from activating the cascade, and there are a number of questions about it.

a) Deletion of ompA changes permeability of the OM (as seen here for entry of the cross-linker, Figure 5—figure supplement 1, and an observation in previous work from this group that the OM becomes permeable to an antibody in the absence of OmpA); does 1-170 or OmpAX restore "wild-type" permeability? This is important in distinguishing effects on the membrane affecting signaling (due to changes in permeability/behavior) from formation of a complex with RcsF that affects signaling.

b) In work by others, deletion of ompA no longer responds to an inducing signal. This figure looks only at basal level; how do these constructs respond to inducing signals?

4) The connection between observations in terms of NMR contacts, crosslinking, and in vivo effects needs to be reinforced. Are there residues in the C-term of OmpA identified by NMR that can be mutated to see if they fail to interact with RcsF (or even a truncation of the C-terminus). Do these lead to rprA induction as the deletion of ompA does? Do they affect response to signaling when introduced into the single copy of ompA? Otherwise, it is hard to know how many of the interactions from either the NMR or cross-linking are relevant to signaling.

Reviewer #3:

Létoquar et al. investigate sites of interaction between OmpA and RcsF, discovering that RcsF interacts mainly with the C-terminal globular domain of OmpA in the periplasm and that this is competitive with the inner membrane protein IgaA. These interactions potentially play important roles in the functioning of RcsF, which is one of *E. coli*'s key cell envelope stress sensor systems. OmpA is an abundant outer membrane protein consisting of two domains, an N-terminal 8-stranded β-barrel in the outer membrane and a globular domain in the periplasm non-covalently bound to the peptidoglycan (PG). Recently, both the Collet and Silhavy groups have shown that porins (OmpF/C and possibly OmpA) are involved in presenting RcsF on the cell surface although the region of RscF that is present is disputed. Here, the major focus is on OmpA-RcsF interactions in vivo and in vitro and how these are modulated by another component of the rcs relay, IgaA. The major issue with the paper is the absence of definitive biochemical and biophysical data supporting the main conclusion that IgaA and OmpA compete for binding to RcsF.

1) A limitation with the paper as presented is that there are no equilibrium dissociation constants presented and no definitive experiments showing competitive binding between IgaA and the PG-binding domain of OmpA. Speculative arguments are made about possible affinities of complexes (Results, Discussion and Materials and methods) rather than direct binding measurements between the different binding partners in vitro. For example, by SPR or ITC. NMR titration data are shown (Figure 4) but no affinities are derived from these nor is it demonstrated that binding saturates in these experiments. NMR could also have been used to show competition with IgaA but was not.

2) The paper nicely narrows down binding between OmpA PG-binding domain and RscF using crosslinking, thrombin sites and OmpX chimera. But does IgaA actually compete for the same site? The only experiment related to competition is in vivo crosslinking in conjunction with overexpression of IgaA (Figure 5). Can the authors discount a physiological response as being responsible for the decrease in OmpA-RcsF crosslinking rather than direct competition?

3) Photoactivated crosslinking was used to identify sites on RcsF that link to OmpA but no LC-MS/MS was carried out to identify the precise sites on OmpA where crosslinking occurs, which is entirely feasible. This would have made the story much clearer. Moreover, if the same could be done with RcsF crosslinking to IgaA then it would be immediately apparent if IgaA shared the same binding site as OmpA. As it is, little can be said about competition other than a suggestion of this from Figure 5.

4) The C-terminal domain of OmpA normally binds PG. Does binding to RcsF block binding to PG? This is a relatively trivial experiment to do (using isolated sacculi for example) and should have been included given the involvement of PG in the cell envelope stress response of RcsF.

5) The authors suggest that the present work supports their argument that OmpA is not used as a conduit for RcsF to reach the surface of *E. coli*. Wouldn't a more direct approach be a cell ELISA using their RcsF antibodies in conjunction with porin deletion mutants with/without OmpA being expressed?

[Editors’ note: further revisions were suggested prior to acceptance, as described below.]

Thank you for submitting your article "Defining the function of OmpA in the Rcs stress response" for consideration by *eLife*. Your article has been reviewed by three peer reviewers, and the evaluation has been overseen by a Reviewing Editor and Gisela Storz as the Senior Editor. The following individual involved in review of your submission has agreed to reveal their identity: Samuel I Miller (Reviewer #2).

The reviewers have discussed the reviews with one another and the Reviewing Editor has drafted this decision to help you prepare a revised submission.

This paper presents strong evidence indicating that RcsF, the sensing lipoprotein of the Rcs phosphorelay system, interacts with the periplasmic globular domain of OmpA rather than through binding of the N-terminus of RcsF to the β-barrel portion of OmpA, as previously suggested by others. Based on these and other recent findings, the authors propose a compelling model in which the RcsF/OmpA interaction in the periplasm helps to buffer the ability of RcsF to bind IgaA and induce the Rcs phoshporelay. Although the reviewers found the revised manuscript to be substantially improved, there were several issues that still need to be addressed, as summarized below. It was agreed that these issues can likely be addressed simply by modifying the text, although the authors should add new data and experiments if they think it would bolster their arguments.

1) The revised manuscript provides significantly improved data on the lack of a required interaction of RcsF with the β-barrel portion of OmpA. It is certainly striking that chimeric constructions that are capable of complementing the higher basal level of Rcs expression seen in cells deleted for RcsA. What is a bit less clear is how the interaction with OmpA affects the response of the Rcs system. The authors show in Figure 6—figure supplement 1 that their ompA deletion is still inducible with polymyxin, with a similar induced level with the deletion. What exactly would the buffering be expected to show? Should it change the dose of Polymyxin, for instance, that would induce? Can that be seen, or only the basal level of repression in the absence of signal?

2) Is there any explanation for how MG1655 and MC4100 might differ for effects in the absence of OmpA?

3) Figure 4: It would be useful to indicate in panel C and D where the OmpA residues that don't interact well with RcsF in panel E sit. They appear to be either absent or very low in terms of CSP in Figure 4B. Why were these chosen? Do mutations in these sites have any phenotype for the Rcs system in vivo? For instance, does deletion or mutation of the flexible loop act like an ompA deletion for basal level expression of the Rcs system? If such a deletion lost interaction, it would be a very useful control throughout this work. Where do those mutations sit in the model of the complex between RcsF and OmpA in Figure 6?

4) Figure 5B: Why is the amount of the BamA-RcsF much lower when IgaA is overexpressed in the absence of OmpA (lane 12)? Does that suggest that OmpA is necessary for RcsF to interact with BamA (rather than the model, which I believe is the other way around)?

5) Figure 5B: The idea that there is a competition for RcsF between IgaA and OmpA derives in significant part from this data, showing a decrease in the OmpA/RcsF complex when IgaA was overexpressed. Can these changes be quantitated?

6) Competitive binding between OmpA and IgaA for RcsF. The data in the paper make a good case for both the periplasmic domain of OmpA and IgaA binding RcsF. However, the case for competitive binding is still not convincing. There is only one in vivo crosslinking blot that addresses this point, as in the original manuscript, in cells overexpressing IgaA (Figure 5B). For one, the authors should quantify these data. Also, the authors make the point that the massively different binding affinities of IgaA and OmpA for RcsF poses technical challenges to assessing the competition model. But would this necessarily be the case if they isotopically labelled RcsF (rather than OmpA) and demonstrated IgaA and OmpA cause the same residues to shift in heteronuclear NMR experiments? If the authors do not provide additional NMR data, they can only safely conclude IgaA and OmpA interact with RcsF. They would be on less solid ground when it comes to saying this is competitive as it's not clear that they can discount an allosteric mechanism. Either new data should be provided to substantiate the claims of a competition model or the text should be carefully revised to indicate that this model has not be definitively shown yet.

7) The new docked model of the RcsF-OmpA complex raises new questions. The complex is described as being “remarkable” but it is not clear why? What are the statistics for this modelled complex to indicate how good a fit it is (e.g. what is the accessible surface area that is buried?) and how well it explains the crosslinking data? Also, it would have been useful to not only discuss the ramifications of this complex for OmpA function (the PG binding region is exposed apparently although this is not obvious from Figure 6) but to also indicate how this relates to RcsF functionality. The group recently published the crystal structure of RcsF bound to the lumen of BamA. Are the residues/regions of RcsF involved in BamA binding also involved in binding to OmpA or are they different?

8) The authors should use reduce their use of the word “remarkable” throughout the manuscript and reserve this and related terms for the 1-2 cases where it is most justified.

---

## [Author Response]

[Editors’ note: the authors resubmitted a revised version of the paper for consideration. What follows is the authors’ response to the first round of review.]

Reviewer #1:In this manuscript the authors find a new function for OmpA in the regulation of the RcsFBC system through a direct interaction with the periplasmic domain of the OmpA protein. Through biochemical and genetic experiments they elegantly establish that the carboxy-terminal domain of OmpA binds and suppresses the activation of RcsF that occurs when OmpA is deleted. The data is of high quality and the work resolves some controversies in the field and should be of general interest.Specific comments:The NMR results confirm that conformational changes in these residues occur when RcsF binds to OmpA, not that these residues are those that are bound directly, and though it is a distinct possibility that these residues form a binding interface the authors should be more cautious in the interpretation of their results.

We fully agree with this reviewer. In the previous version of the manuscript, we used the Cterminal domain of *Klebsiella pneumoniae* OmpA (*Kp*OmpA), a protein with 88.49% sequence identity with its *E. coli* counterpart. To take this comment into account, we decided to redo and extend the NMR analysis using the periplasmic domain of *E. coli* OmpA. This allowed us to 1) identify a flexible loop located at the tip of the periplasmic domain of OmpA as part of the binding interface with RcsF (new Figure 4 A, B, C, D) and 2) to calculate the equilibrium dissociation constant (*K*_D_) of OmpA for RcsF as being ~100 µM (new Figure 5B). In addition, using site-specific photo-crosslinking, we confirmed that residues from the flexible loop identified by NMR directly bind RcsF (new Figure 4E).

The authors should be cautious about comparing the affinity of the interaction between IgaA and RcsF versus that between OmpA and RcsF based on the results with copy number, as many other factors include protein folding and accessary proteins/peptidoglycan binding could play a role in vivo.

This reviewer is right. To address her/his comment, we determined the equilibrium dissociation constants of the periplasmic domains of IgaA and OmpA for RcsF (new Figure 5 B,C). Remarkably, our new results confirm our prediction that IgaA has substantially more affinity for RcsF than OmpA (~1.6 nM versus 125 µM). In addition, we added in the Discussion that the K_D_ values measured in vitro are likely to be affected by different factors in vivo, as mentioned by this reviewer.

The authors acutely point out the fact that it is hard to separate the effects of Bam binding to RcsF from its role in folding OmpA. However, does a periplasmic expressed carboxyl-terminal domain of OmpA (made as a fusion with a cleaved signal sequence) suppress the activation of RcsF in a OmpA null mutant, and if it does then a Bam null mutation could be added and would likely determine no effect on suppression. This experiment would rule out the rare possibility that the OmpX fusion has some unexpected binding through the OmpX domain.

To address this comment, we expressed the C-terminal domain of OmpA as an outer membrane lipoprotein (OmpA_Pal_) by fusing it to the signal sequence and lipobox (for lipid modification) of the lipoprotein Pal. We found that OmpA_Pal_ binds RcsF and that its expression decreases Rcs activation (new Figure 3C). Thus, these new results rule out the possibility that the suppression observed when OmpA_X_ is expressed results from some unexpected binding to OmpX.

A BAM null mutation could not be added because BAM is essential.

Reviewer #2:The outer-membrane linked lipoprotein RcsF is necessary for many signals that turn on the Rcs phosphorelay. The current model for signaling suggests that RcsF, when "activated" by an OM or PG stress, interacts with IgaA, relieving IgaA repression of Rcs signaling. How activation occurs, and in what way RcsF might be sequestered in the absence of signal and released upon signalling is the focus of the work here and would be of general interest in understanding this complex phosphorelay and its unique signaling mechanism. Previous findings had shown that parts of RcsF were exposed on the cell surface, and that RcsF interacted with outer membrane proteins, including OmpA, leading to a suggestion that RcsF might be threaded through the center of some porins (not necessarily but possibly OmpA) to the cell surface. This paper uses cross-linking and some chimeric protein constructs to suggest that RcsF binds to the periplasmic, rather than the β-barrel domain of OmpA, and that this binding acts to sequester RcsF. There are some major concerns that compromise the impact of this work: Many of the experiments require additional controls to confirm that they are assaying what they are said to assay, and the relationship of these findings to the critical in vivo signaling pathway is not currently well supported. Overall, it is not clear that the new information provided here adds significantly to our understanding of how RcsF signals; it may rule out some possibilities.

We thank the reviewer for carefully evaluating our manuscript and for her/his insightful and constructive comments. We do not think, however, that the main focus of our work is understanding how Rcs activation occurs and how RcsF may be released upon signaling. In fact, we recently published an article reporting the structure of a complex between BamA and RcsF in which we propose, based on our structural data, a mechanism for Rcs activation under outer membrane stress. In this mechanism, perturbing the conformational cycling that occurs in BamA during b-barrel insertion is key to allow RcsF to trigger Rcs (Rodriguez-Alonso et al., 2020).

In the current manuscript, we focus instead on the OmpA-RcsF complex in order to determine if OmpA functions as a vehicle allowing RcsF to reach the surface or not. This information is crucial to understand the role played by OmpA in the complex Rcs system. We respectfully disagree with this reviewer when she/he writes that “RcsF might be threaded through the center of some porins (not necessarily but possibly OmpA)”. We cannot downplay the role of OmpA because it was previously reported that RcsF uses its surface-exposed N-terminal linker to detect stress on the basis of experiments focusing exclusively on the OmpA-RcsF complex (Konovalova et al., 2016).

It is therefore crucial to determine whether or not OmpA, which unlike OmpC and OmpF, folds into a small, 8-stranded b-barrel with no central cavity, allows RcsF to reach the cell surface. The information that we report in the current investigation provides a solid foundation for understanding the complex Rcs system: our in vivo and in vitro data clearly show that RcsF interacts with the periplasmic domain of OmpA and that the b-barrel domain is not involved (see our responses below for more details). These results indicate that OmpA does not mediate the surface exposure of RcsF, as proposed by us and by others (Cho et al., 2014; Konovalova et al., 2014), and implies that the OmpA-RcsF complex does not serve to monitor the state of the outer membrane via direct interactions between RcsF and lipopolysaccharide molecules, as previously reported (Konovalova et al., 2016). Our results reveal instead that OmpA functions as a buffer regulating Rcs activity by competing with IgaA for RcsF binding.

We adapted the Abstract, the Introduction and the Discussion sections to better highlight the novelty, importance and general interest of our results and to better discuss the key conceptual message, i.e. that OmpA does not mediate the surface exposure of RcsF but rather regulates Rcs signaling.

1) Figure 1: Site-specific crosslinking of OmpA and RcsF: RcsF is expressed from a high copy plasmid, and OmpA is very abundant. What is the evidence for site-specific cross-linking? At a minimum, does wild-type RcsF without the DiZPK fail to bring down any OmpA when subjected to UV and immunoprecipitation. It would seem a comparison of amounts in such a control would help to support that the cross-linking is due to the cross-linker (and is thus site specific).

As requested by this reviewer, we now show that wild-type RcsF without DiZPK fails to bring down OmpA (new Figure 1). In addition, the fact that introduction of DiZPK at R21 of RcsF (new Figure 1) does not lead to the formation of an OmpA-RcsF complex following UV exposure further confirms the specificity of the technique.

2) In the Figure 1—figure supplement 1: Some of these are said to be positive (Q28, Q121 (mislisted in text as N121) but show very low amounts in upper panel (with RcsF), and some at bottom have very low OmpA signals. Why are these not in the same ratios? Shouldn't they be the same complex?

We agree with this reviewer that the amounts of the OmpA-RcsF complex that were detected depended on the antibodies (anti-RcsF or anti-OmpA) that were used. To address this problem, the experiment was repeated in triplicates with an optimized cross-linking and identification protocol. Two new figures (new Figure 1 and new Figure 1—figure supplement 1) were prepared. In new Figure 1, we observe the UV-dependent formation of the OmpA-RcsF complex as detected by anti-RcsF antibodies. In new Figure 1—figure supplement 1, we show that the UV-dependent ~50 kDa band is not detected by the anti-RcsF antibodies when the cross-linking experiment is carried out in a ∆*ompA* mutant, which unambiguously confirms the identification of this band as the OmpA-RcsF complex and solves the problem of using different antibodies.

3) Figure 3D: This in vivo experiment is a critical part of the argument that only the periplasmic region of OmpA is critical for keeping RcsF from activating the cascade, and there are a number of questions about it.a) Deletion of ompA changes permeability of the OM (as seen here for entry of the cross-linker, Figure 5—figure supplement 1, and an observation in previous work from this group that the OM becomes permeable to an antibody in the absence of OmpA); does 1-170 or OmpAX restore "wild-type" permeability? This is important in distinguishing effects on the membrane affecting signaling (due to changes in permeability/behavior) from formation of a complex with RcsF that affects signaling.

We understand the concern of this reviewer. Expression of OmpA_X_ restores almost wild-type OM permeability in the ∆*ompA* mutant (not shown), which does not allow us to distinguish between “effects on the membrane affecting signaling from formation of a complex that affects signaling”. Thus, we carried out a novel experiment in which we expressed the periplasmic domain of OmpA as an outer membrane lipoprotein (OmpA_Pal_) by fusing it to the signal sequence and lipobox of the lipoprotein Pal. OmpA_Pal_ does not possess an N-terminal b-barrel that could stabilize the OM, restoring wild-type permeability. Remarkably, we found that OmpA_Pal_ interacts with RcsF (new Figure 3C) and that its expression substantially decreases Rcs activity (new Figure 3C). Thus, the results obtained with OmpA_Pal_ provide direct evidence 1) for the formation of a complex between RcsF and the periplasmic domain of OmpA and 2) for the fact that the periplasmic domain of OmpA keeps RcsF from activating the cascade.

b) In work by others, deletion of ompA no longer responds to an inducing signal. This figure looks only at basal level; how do these constructs respond to inducing signals?

We found that exposure to polymyxin B induces Rcs activity even in cells lacking OmpA (new Figure 6—figure supplement 1). As explained in the Discussion, the discrepancy could come from the fact that we used DH300, a MG1655 derivative, and that Konovalova et al. used the strain MC4100.

4) The connection between observations in terms of NMR contacts, crosslinking, and in vivo effects needs to be reinforced. Are there residues in the C-term of OmpA identified by NMR that can be mutated to see if they fail to interact with RcsF (or even a truncation of the C-terminus). Do these lead to rprA induction as the deletion of ompA does? Do they affect response to signaling when introduced into the single copy of ompA? Otherwise, it is hard to know how many of the interactions from either the NMR or cross-linking are relevant to signaling.

The following new experiments were done to take this comment into account and improve the connection between the in vivo and in vitro observations:

1) We redid and extended the NMR analysis using the periplasmic domain of *E. coli* OmpA and not the *Klebsiella pneumoniae* homolog like in the previous version. This allowed us to 1) identify a flexible loop located at the tip of the periplasmic domain of OmpA as part of the binding interface with RcsF (new Figure 4 A, B, C, D, E) and 2) to calculate the equilibrium dissociation constant (*K*_D_) of OmpA for RcsF as being ~125 µM (new Figure 5B).

2) We carried out additional site-specific crosslinking experiments to confirm the results from the new NMR experiments and to identify OmpA residues involved in the interaction with RcsF (see our response to comment #1 from reviewer 1).

3) We used protein-protein docking and molecular dynamics simulations to build a threedimensional model taking into accounts the results of the cross-linking experiments; in this model, RcsF and the periplasmic domain of OmpA show a remarkable surface complementarity.

Altogether, the new results provide crucial information on the formation and properties of the OmpA-RcsF complex and reinforce the connection between the in vivo and in vitro investigations.

To satisfy this reviewer, we also generated a number of OmpA variants with single mutations in the periplasmic domain but all of them kept the ability to interact with RcsF. It is likely that variants with multiple mutations need to be engineered to lose the interaction with RcsF. However, in this case, the overall conformation of OmpA will likely be affected, which will prevent clear interpretation of the results.

Reviewer #3:Létoquar et al. investigate sites of interaction between OmpA and RcsF, discovering that RcsF interacts mainly with the C-terminal globular domain of OmpA in the periplasm and that this is competitive with the inner membrane protein IgaA. These interactions potentially play important roles in the functioning of RcsF, which is one of E. coli's key cell envelope stress sensor systems. OmpA is an abundant outer membrane protein consisting of two domains, an N-terminal 8-stranded β-barrel in the outer membrane and a globular domain in the periplasm non-covalently bound to the peptidoglycan (PG). Recently, both the Collet and Silhavy groups have shown that porins (OmpF/C and possibly OmpA) are involved in presenting RcsF on the cell surface although the region of RscF that is present is disputed. Here, the major focus is on OmpA-RcsF interactions in vivo and in vitro and how these are modulated by another component of the rcs relay, IgaA. The major issue with the paper is the absence of definitive biochemical and biophysical data supporting the main conclusion that IgaA and OmpA compete for binding to RcsF.1) A limitation with the paper as presented is that there are no equilibrium dissociation constants presented and no definitive experiments showing competitive binding between IgaA and the PG-binding domain of OmpA. Speculative arguments are made about possible affinities of complexes (Results, Discussion and Materials and methods) rather than direct binding measurements between the different binding partners in vitro. For example, by SPR or ITC. NMR titration data are shown (Figure 4) but no affinities are derived from these nor is it demonstrated that binding saturates in these experiments. NMR could also have been used to show competition with IgaA but was not.

The equilibrium dissociation constants of the periplasmic domains of OmpA and IgaA for RcsF have now been determined using new NMR titration experiments and biolayer interferometry, respectively (new Figure 5 C, D). The extremely high affinity that IgaA has for RcsF in vitro (*K*_D_=~1 nM) prevented us to carry out competition experiments with the periplasmic domain of OmpA (*K*_D_=~125 µM).

2) The paper nicely narrows down binding between OmpA PG-binding domain and RscF using crosslinking, thrombin sites and OmpX chimera. But does IgaA actually compete for the same site?

We agree with this reviewer that defining the binding interface of IgaA for RcsF is very interesting. We are currently trying to crystallize the IgaA-RcsF complex in order to solve its structure. However, we think that addressing this question goes beyond the scope of the current investigation.

The only experiment related to competition is in vivo crosslinking in conjunction with overexpression of IgaA (Figure 5). Can the authors discount a physiological response as being responsible for the decrease in OmpA-RcsF crosslinking rather than direct competition?

As far as we know, IgaA is dedicated to the regulation of the Rcs system. In the competition experiment, we verified that IgaA over-expression did not induce Rcs, which could have impacted the results. In addition, we show that the amounts of OmpA and of RcsF do not change when IgaA is overexpressed (Figure 5B) and that the levels of a complex that OmpA forms with an unknown protein (Figure 5B, lower panel) also remain unchanged. We therefore do not think that the decrease in OmpA-RcsF results from a physiological response. The newly determined *K*_D_ values also agree well with the idea that IgaA and OmpA both compete for RcsF.

3) Photoactivated crosslinking was used to identify sites on RcsF that link to OmpA but no LC-MS/MS was carried out to identify the precise sites on OmpA where crosslinking occurs, which is entirely feasible. This would have made the story much clearer. Moreover, if the same could be done with RcsF crosslinking to IgaA then it would be immediately apparent if IgaA shared the same binding site as OmpA. As it is, little can be said about competition other than a suggestion of this from Figure 5.

We tried to identify the RcsF-OmpA crosslinks using LC-MS/MS, but failed. In our study, we used DiZPK instead of pBPA for cross-linking, which makes the analysis of the MS/MS spectra substantially more complicated. However, using site-specific photocrosslinking, we identified OmpA residues that interact with RcsF (new Figure 4E). See also our responses to similar comments made by reviewers 1 and 2.

4) The C-terminal domain of OmpA normally binds PG. Does binding to RcsF block binding to PG? This is a relatively trivial experiment to do (using isolated sacculi for example) and should have been included given the involvement of PG in the cell envelope stress response of RcsF.

We agree with this reviewer that it would be very interesting to study the impact of PG-binding by OmpA on complex formation with RcsF. In the new manuscript, we used protein-protein docking and molecular dynamics simulations to build a three-dimensional model taking into accounts the results of the cross-linking experiments (new Figure 6); in this model, the peptidoglycan-binding region of OmpA remains accessible, which suggests that RcsF binding does not prevent OmpA from interacting with the PG. However, we think that a detailed investigation of this question is beyond the scope of the current manuscript.

5) The authors suggest that the present work supports their argument that OmpA is not used as a conduit for RcsF to reach the surface of E. coli. Wouldn't a more direct approach be a cell ELISA using their RcsF antibodies in conjunction with porin deletion mutants with/without OmpA being expressed?

We cannot use antibodies-based approaches such as ELISA to study the impact of deleting *ompA* on the surface exposure of RcsF because this deletion increases the permeability of the outer membrane to antibodies, as reported previously (Cho et al., 2014).

[Editors’ note: what follows is the authors’ response to the second round of review.]

This paper presents strong evidence indicating that RcsF, the sensing lipoprotein of the Rcs phosphorelay system, interacts with the periplasmic globular domain of OmpA rather than through binding of the N-terminus of RcsF to the β-barrel portion of OmpA, as previously suggested by others. Based on these and other recent findings, the authors propose a compelling model in which the RcsF/OmpA interaction in the periplasm helps to buffer the ability of RcsF to bind IgaA and induce the Rcs phoshporelay. Although the reviewers found the revised manuscript to be substantially improved, there were several issues that still need to be addressed, as summarized below. It was agreed that these issues can likely be addressed simply by modifying the text, although the authors should add new data and experiments if they think it would bolster their arguments.

We thank the reviewers and the editors for carefully evaluating our manuscript and providing insightful comments. We are glad that they found the new version of the manuscript to be substantially improved. We carried out new experiments to address their questions and modified the manuscript as requested.

1) The revised manuscript provides significantly improved data on the lack of a required interaction of RcsF with the β-barrel portion of OmpA. It is certainly striking that chimeric constructions that are capable of complementing the higher basal level of Rcs expression seen in cells deleted for RcsA. What is a bit less clear is how the interaction with OmpA affects the response of the Rcs system. The authors show in Figure 6—figure supplement 1 that their ompA deletion is still inducible with polymyxin, with a similar induced level with the deletion. What exactly would the buffering be expected to show? Should it change the dose of Polymyxin, for instance, that would induce? Can that be seen, or only the basal level of repression in the absence of signal?

This is an excellent question. We think that OmpA functions as a buffer for RcsF under normal or mild stress conditions, not when the envelope is severely damaged (severe disruption of the outer membrane will impact the OmpA-RcsF complex). In the experiment that was shown in Figure 6—figure supplement 1, polymyxin B was used at a concentration of 0.5 µg/ml. This concentration is well below the MIC (see our response to comment #2), but fully induces Rcs, which is indicative of relatively severe outer membrane stress. To address the question raised by the reviewers, we carried out new experiments to probe Rcs induction by lower concentrations of polymyxin B (0.0001, 0.001 and 0.01 µg/ml). As shown in the new Figure 6—figure supplement 1, we see higher Rcs induction in the ∆*ompA* mutant than in the wild type, which illustrates the buffering function of OmpA. The manuscript has been modified accordingly (subsection “OmpA functions as a buffer for RcsF”).

2) Is there any explanation for how MG1655 and MC4100 might differ for effects in the absence of OmpA?

There are major differences between MG1655 and MC4100: comparison of the genome of MC4100 to that of MG1655 revealed the deletion of dozens of open reading frames (Peters, Thate et al., J. Bact., 2003) as well as the presence of regulatory mutations and recombinational events (Ferenci, Zhou et al., J. Bact., 2009).

In an attempt to find an explanation to the different behavior of the two strains, we determined the MIC of polymyxin B for MC4100 and MG1655, but found that they are similar (2 µg/ml). We also tested Rcs induction by polymyxin B in MC4100 and found that Rcs is induced, also in cells lacking OmpA (see Author response image 1). We think that discussing this further could distract the reader from the main message of the paper; we therefore prefer not to include this graph in the manuscript.

**Author response image 1. sa2fig1:** Rcs induction by polymyxin B in MC4100 and MG1655. Wild-type and ∆ompA cells were treated with polymyxin B (PMB; 0.5 µg/ml) when they reached an OD600 of 0.4. Rcs activity was measured by a chromosomal rprA::lacZ fusion. Treatment with polymyxin B activated Rcs in the wild type and in the ∆ompA mutant, both in MG1655 and MC4100. Mean (n=3) and standard deviation (error bars) are shown. Differences were evaluated with Student’s t test (ns, not significant; * p<0.05, *** p<0,001).

3) Figure 4: It would be useful to indicate in panel C and D where the OmpA residues that don't interact well with RcsF in panel E sit. They appear to be either absent or very low in terms of CSP in Figure 4B. Why were these chosen? Do mutations in these sites have any phenotype for the Rcs system in vivo? For instance, does deletion or mutation of the flexible loop act like an ompA deletion for basal level expression of the Rcs system? If such a deletion lost interaction, it would be a very useful control throughout this work. Where do those mutations sit in the model of the complex between RcsF and OmpA in Figure 6?

Figure 4 and Figure 6 and their legends have been modified as requested by the reviewers. R242 and Y248 were chosen because they are in the flexible loop. Also see our response to comment #7.

We tested the following mutations in the C-terminal domain of OmpA: H193E, N205E, K206E, K206W, R242E, S245W, D246W, Y248E, Q250W, N288K, K192A, K294E. We also prepared double and triple mutants by combining the previously mentioned mutations. None of the tested mutant lost interaction with RcsF. They also did not induce Rcs.

We did not delete the flexible loop. However, when the sequence of a thrombin cleavage site was introduced after residue 243 (in the loop), we could still observe the OmpA-RcsF interaction, as shown in the manuscript (Figure 3).

4) Figure 5B: Why is the amount of the BamA-RcsF much lower when IgaA is overexpressed in the absence of OmpA (lane 12)? Does that suggest that OmpA is necessary for RcsF to interact with BamA (rather than the model, which I believe is the other way around)?

The ∆*ompA* strain was used as a negative control in the experiment shown in Figure 5B. The absence of OmpA in the outer membrane changes the permeability of this membrane, as shown by us (Cho et al., 2014) and by others (Paradis-Bleau et al., PLOS Genetics, 2014). Therefore, we cannot compare the intensity of the crosslinked complexes between the wild type and the ∆*ompA* mutant because the permeability of the outer membrane to the chemical crosslinker BS3 is different. We would expect to see more crosslinking in the ∆*ompA* strain because deletion of *ompA* makes the cells more permeable, but we see the opposite (Figure 5B). We do not know the reason. Importantly, we observed similar amounts of OmpA crosslinks (bottom panel in Figure 5B) in wild-type cells expressing increasing amounts of IgaA as well as in the ∆*rcsF* mutant, which indicates that the permeability of the outer membrane to BS3 is similar in these cells.

5) Figure 5B: The idea that there is a competition for RcsF between IgaA and OmpA derives in significant part from this data, showing a decrease in the OmpA/RcsF complex when IgaA was overexpressed. Can these changes be quantitated?

Figure 5B has been modified to include a panel showing the quantitation. The legend has been modified accordingly.

6) Competitive binding between OmpA and IgaA for RcsF. The data in the paper make a good case for both the periplasmic domain of OmpA and IgaA binding RcsF. However, the case for competitive binding is still not convincing. There is only one in vivo crosslinking blot that addresses this point, as in the original manuscript, in cells overexpressing IgaA (Figure 5B). For one, the authors should quantify these data. Also, the authors make the point that the massively different binding affinities of IgaA and OmpA for RcsF poses technical challenges to assessing the competition model. But would this necessarily be the case if they isotopically labelled RcsF (rather than OmpA) and demonstrated IgaA and OmpA cause the same residues to shift in heteronuclear NMR experiments? If the authors do not provide additional NMR data, they can only safely conclude IgaA and OmpA interact with RcsF. They would be on less solid ground when it comes to saying this is competitive as it's not clear that they can discount an allosteric mechanism. Either new data should be provided to substantiate the claims of a competition model or the text should be carefully revised to indicate that this model has not be definitively shown yet.

We could not perform competition experiments using NMR. We therefore modified the text as requested by the reviewers to tone down our statements (including in the Abstract) that there is a competitive binding between OmpA and IgaA for RcsF.

7) The new docked model of the RcsF-OmpA complex raises new questions. The complex is described as being “remarkable” but it is not clear why? What are the statistics for this modelled complex to indicate how good a fit it is (e.g. what is the accessible surface area that is buried?) and how well it explains the crosslinking data? Also, it would have been useful to not only discuss the ramifications of this complex for OmpA function (the PG binding region is exposed apparently although this is not obvious from Figure 6) but to also indicate how this relates to RcsF functionality. The group recently published the crystal structure of RcsF bound to the lumen of BamA. Are the residues/regions of RcsF involved in BamA binding also involved in binding to OmpA or are they different?

We calculated the buried accessible surface area using the SASA module of GROMACS (Abraham et al., 2015). We found it to be 1087 Å^2^, which represents 9% and 7% of the total protein surface area of RcsF and OmpA (periplasmic domain), respectively. The proposed interaction model is in good agreement with the crosslinking data, with all six residues mutated to DiZPK (RcsF_R45X_, RcsF_Q79X_, RcsF_R89X_, OmpA_R242X_, OmpA_D246X_, and OmpA_Y248X_) situated in close proximity to residues from the binding partner (new Figure 6A). This information has been added to the manuscript.

We also modified the manuscript to explain that several residues of RcsF found to interact with OmpA (Q79 – R89 – P116) are part of the binding interface between RcsF and the luminal wall of the BamA β-barrel in our recently published structure of BamA-RcsF (Rodriguez-Alonso et al., 2020).

8) The authors should use reduce their use of the word “remarkable” throughout the manuscript and reserve this and related terms for the 1-2 cases where it is most justified.

The manuscript was modified accordingly.